# Unbiased proteomics, histochemistry, and mitochondrial DNA copy number reveal better mitochondrial health in muscle of high-functioning octogenarians

Ceereena Ubaida-Mohien[1], Sally Spendiff[2], Alexey Lyashkov[1], Ruin Moaddel[1], Norah J MacMillan[3], Marie-Eve Filion[3], Jose A Morais[3], Tanja Taivassalo[4], Luigi Ferrucci[1]*‡, Russell T Hepple[4,5]*‡

[1]Intramural Research Program, National Institute on Aging, National Institutes of Health, Baltimore, United States; [2]Research Institute, Children's Hospital of Eastern Ontario, Ottawa, Canada; [3]Research Institute of the McGill University Health Centre, McGill University, Montreal, Canada; [4]Department of Physical Therapy, University of Florida, Gainesville, United States; [5]Department of Physiology and Functional Genomics, University of Florida, Gainesville, United States

*For correspondence:
ferruccilu@grc.nia.nih.gov (LF);
rthepple@ufl.edu (RTH)

‡These authors shared senior authorship to this work

Competing interest: The authors declare that no competing interests exist.

## Abstract

**Background:** Master athletes (MAs) prove that preserving a high level of physical function up to very late in life is possible, but the mechanisms responsible for their high function remain unclear.

**Methods:** We performed muscle biopsies in 15 octogenarian world-class track and field MAs and 14 non-athlete age/sex-matched controls (NA) to provide insights into mechanisms for preserving function in advanced age. Muscle samples were assessed for respiratory compromised fibers, mitochondrial DNA (mtDNA) copy number, and proteomics by liquid-chromatography mass spectrometry.

**Results:** MA exhibited markedly better performance on clinical function tests and greater cross-sectional area of the vastus lateralis muscle. Proteomics analysis revealed marked differences, where most of the ~800 differentially represented proteins in MA versus NA pertained to mitochondria structure/function such as electron transport capacity (ETC), cristae formation, mitochondrial biogenesis, and mtDNA-encoded proteins. In contrast, proteins from the spliceosome complex and nuclear pore were downregulated in MA. Consistent with proteomics data, MA had fewer respiratory compromised fibers, higher mtDNA copy number, and an increased protein ratio of the cristae-bound ETC subunits relative to the outer mitochondrial membrane protein voltage-dependent anion channel. There was a substantial overlap of proteins overrepresented in MA versus NA with proteins that decline with aging and that are higher in physically active than sedentary individuals. However, we also found 176 proteins related to mitochondria that are uniquely differentially expressed in MA.

**Conclusions:** We conclude that high function in advanced age is associated with preserving mitochondrial structure/function proteins, with underrepresentation of proteins involved in the spliceosome and nuclear pore complex. Whereas many of these differences in MA appear related to their physical activity habits, others may reflect unique biological (e.g., gene, environment) mechanisms that preserve muscle integrity and function with aging.

**Funding:** Funding for this study was provided by operating grants from the Canadian Institutes of Health Research (MOP 84408 to TT and MOP 125986 to RTH). This work was supported in part by the Intramural Research Program of the National Institute on Aging, NIH, Baltimore, MD, USA.

## Editor's evaluation

Proteomics studies of skeletal muscle biopsies in healthy individuals demonstrate that older age was associated with an underrepresentation of mitochondrial proteins, especially those associated with oxidative phosphorylation and energy metabolism. Ubaida-Mohien et al. analyzed muscle protein differences between octogenarian master athletes and non-athletes demonstrating that high muscle function during aging is associated with the preservation of structural and functional proteins in mitochondria such as electron transport capacity, cristae formation, mitochondrial biogenesis, and mtDNA-encoded proteins. The authors propose that the study of these unique proteins may uncover molecular mechanisms to design therapeutic strategies for skeletal muscle functional decline with aging.

## Introduction

The aging process is associated with profound changes in body composition that includes a substantial decline of muscle mass and a disproportionally more severe decline in strength (*Goodpaster et al., 2006*). Such decline in skeletal muscle mass and strength starts between the third and the fourth decades of life both in men and women, substantially accelerates after the age of 75 years, and in some individuals becomes so severe as to cause mobility loss and frailty (*Cawthon et al., 2020*). However, there is clear evidence that the degree of such 'usual' decline of strength and function is less severe in some individuals. For example, master athletes (MAs) exhibit considerably higher physical performance capacity in their 80s and 90s than their sedentary counterparts and there have been sporadic mentions of centenarians who compete in marathons (https://www.runnersworld.com/runners-stories/a20812407/whos-the-fastest-centenarian/). The study of these extreme examples provides a unique opportunity to identify mechanisms that in most individuals determine a decline of muscle health with aging, but that are partially counteracted in highly functioning individuals. For example, we have previously shown in a cohort of highly functioning octogenarian track and field athletes that there was better maintenance of the number and transmission stability of motor units (*Power et al., 2016*) and indications of high muscle fiber reinnervation capacity (*Sonjak et al., 2019*) compared to healthy octogenarian non-athletes and pre-frail/frail octogenarians, respectively.

Using an unbiased discovery proteomics approach on skeletal muscle biopsies collected in very healthy individuals aged 20–87 years, we previously found that older age was associated with underrepresentation of mitochondrial proteins, especially those associated with oxidative phosphorylation (OXPHOS) and energy metabolism (*Ubaida-Mohien et al., 2019b*). Besides, independent of age, 75% of proteins overrepresented in persons who were more physically active in their daily life were mitochondrial proteins across the different sub-localization or function (*Ubaida-Mohien et al., 2019a*). These data strongly suggest that maintaining mitochondrial function is a key to healthy muscle with aging. However, because both mitochondrial function and physical activity level decline with aging even in healthy individuals, discriminating their independent effects on muscle health remains problematic. The study of muscle biopsies in highly trained, older individuals compared with age-matched controls should overcome, at least in part, this limitation.

In this study, we used data and biological specimens collected in 15 track and field MAs aged 75–93 years (eight females), 8 of whom were world record holders in their age group for at least one event at the time of study, with the remaining individuals ranked in the top five worldwide for their respective age and discipline. These individuals are representative of the extreme tail of the distribution of physical fitness in their age group. These MAs were compared with 14 age- and sex-matched non-athletes recruited from the greater Montreal area (NA; six females) to represent healthy independent octogenarian individuals. We compared in these two groups cardiopulmonary fitness (cycle test), isokinetic knee extensor strength, and lower extremity function (time to walk 4 m fast, chair stands, timed-up and go fast, balance time). In addition, we used MRI of the thigh to determine muscle cross-sectional area (CSA). We performed in-depth skeletal muscle phenotyping using muscle biopsies collected by Bergstrom needle from the vastus lateralis for an unbiased proteomics analyses, histochemical characterization of proteins involved in OXPHOS, and assessment of mitochondrial DNA (mtDNA) copy number by real-time polymerase chain reaction (qPCR). High physical function in octogenarians was associated with overrepresentation of the mitochondrial proteome, underrepresentation of mRNA processing and pre-mRNA splicing, fewer OXPHOS compromised muscle fibers, and higher mtDNA

copy number, implicating mitochondrial health in skeletal muscle as a key feature facilitating high physical function in advanced age.

## Methods

Note that additional details of the methods may be found in Appendix 1.

### Ethical approval

All procedures carried out with human subjects were done with prior approval from the Institutional Review Board of the Faculty of Medicine at McGill University (A08-M66-12B) and according to the Declaration of Helsinki. All subjects provided written informed consent.

### Human subject characteristics

Age- and sex-matched octogenarian world-class track and field athletes (n = 15; eight females) and non-athlete participants (n = 14, six females) were recruited for this study. No explicit power analysis was performed a priori due to the rare nature of the octogenarian world-class athletes, but the premise was to select populations of widely different physical function in advanced age so that insights concerning the role of potential differences in muscle biology in the differences in physical function might be obtained.

### Sample collection

A variety of clinical function tests, thigh CSA by MRI, and a vastus lateralis muscle biopsy were performed in 15 octogenarian world-class track and field athletes and 14 non-athlete age- and sex-matched non-athlete controls. A portion of muscle from a subset of 12 MAs (MA mean age 81.19 ± 5.1 years) and 12 non-athlete controls (NA mean age 80.94 ± 4.5 years) was used from these subjects for liquid-chromatography mass spectrometry (LC-MS) to generate quantitative tandem mass tag (TMT) proteomics data. In addition, we measured mtDNA copy number, the abundance of representative subunits of OXPHOS complexes by Western blot, and muscle histological assessment for fiber type and respiratory compromised fibers (see below).

### Muscle fiber-type labeling and imaging in muscle cross-sections

10-µm-thick sections that were serial to those used in histochemical labeling for respiratory compromised fibers were used in immunolabeling experiments to demonstrate fiber type by probing for the major myosin heavy chain (MHC) isoforms in human skeletal muscle. Sections were first hydrated with 1× phosphate buffered saline (PBS) and blocked with 10% normal goat serum for 30 min in 1× PBS. Sections were subsequently incubated with the following primary antibodies for 1 hr at room temperature: polyclonal rabbit anti-laminin IgG (L9393, 1:700; Sigma-Aldrich), monoclonal mouse anti-MHCI IgG2b (BA-F8, 1:25), monoclonal mouse anti-MHCIIa IgG1 (Sc71, 1:200), and monoclonal mouse anti-MHCIIx IgM (6H1, 1:25). MHC primary antibodies were obtained from the Developmental Studies Hybridoma Bank (University of Iowa, USA). Tissue sections then underwent three washes in 1× PBS, and subsequent incubation with the following secondary antibodies for 1 hr at room temperature: Alex Fluor 488 goat anti-rabbit IgG (A11008, 1:500), Alexa Fluor 350 goat anti-mouse IgG2b (A21140, 1:500), Alex Fluor 594 goat anti-mouse IgG (A21125, 1:500), and Alexa Fluor 488 goat anti-mouse IgM (A21042, 1:500).

Following immunolabeling experiments, slides were imaged with a Zeiss Axio Imager M2 fluorescence microscope (Carl Zeiss, Germany) and analyzed with ImageJ (National Institutes of Health, USA) by an observer blinded to the identity of the samples. An average of 366 ± 131 fibers were analyzed per sample.

### Sample preparation and protein extraction for MS

Roughly, 5–8 mg of vastus lateralis muscle tissue per subject was pulverized in liquid nitrogen and mixed with the modified SDT lysis buffer (100 mM Tris, 140 mM NaCl, 4% SDS, 1% Triton X-114, pH 7.6; Sigma) (*Wiśniewski et al., 2009*). Tissues were sonicated, protein concentration was determined, and the sample quality was confirmed using NuPAGE. 300 µg of muscle tissue lysate was used for tryptic digestion. Samples were basic reverse-phase fractionated and analyzed in nano LCMS/

MS (Q-Exactive HF) using previously published method (*Ubaida-Mohien et al., 2019b*). The method details are given in Appendix 1.

## Proteomics informatics

The raw MS data acquired from 24 samples (MA = 12, NA = 12) was converted to .mgf files (using MSConvert, ProteoWizard 3.0.6002) for each sample fraction and was searched with Mascot 2.4.1 and X!Tandem CYCLONE (2010.12.01.1) using the SwissProt Human sequences from UniProt (version year 2017, 20,200 sequences, appended with 115 contaminants) database. The search engine was set with the following search parameters: TMT 10-plex lysine and n-terminus as fixed modifications and variable modifications of carbamidomethyl cysteine, deamidation of asparagine and glutamate, carbamylation of lysine and n-terminus, and oxidized methionine. A peptide mass tolerance of 20 ppm and 0.08 Da, respectively, and two missed cleavages were allowed for precursor and fragment ions in agreement with the instrument's known mass accuracy. Mascot and X!Tandem search engine results were analyzed in Scaffold Q+ 4.4.6 (Proteome Software, Inc). The TMT channels' isotopic purity was corrected according to the TMT kit. Peptide and protein probability was calculated with PeptideProphet and ProteinProphet probability model (further details are given in Appendix 1).

The log2 transformed reporter ion abundance was normalized by median subtraction from all reporter ion intensity spectra belonging to a protein across all channels. Relative protein abundance was estimated by the median of all peptides for a protein combined. Protein sample loading effects from sample preparations were corrected by median polishing, that is, subtracting the channel median from the relative abundance estimate across all channels to have a median zero as described elsewhere (*Herbrich et al., 2013*; *Kammers et al., 2015*). Quantified proteins were annotated, and corresponding gene names were assigned to each protein for simplicity and data representation. Annotation of the proteins was performed by manual curation and combining information from UniProt, GO, and Reactome database. Further bioinformatics analysis was performed using R programming language (3.4.3) and the free libraries available on Bioconductor. The validation of the age effects and physical activity was performed by comparing the MA dataset with the GESTALT dataset. The details of the GESTALT dataset are available on PRIDE repository PXD011967, and GESTALT subject characteristics are provided in *Figure 5—source data 1*.

## Histochemical labeling for respiratory compromised muscle fibers

COX/SDH histochemistry (*Old and Johnson, 1989*; *Taylor et al., 2003*) was performed to assess the relative activity of OXPHOS complexes IV (COX) and II (SDH), and thus identify muscle fibers with low (Cox^Int) or deficient (COX^Neg) COX activity relative to SDH activity. The COX incubation medium was prepared by adding 100 µM cytochrome *c* to 4 mM of 3,3-diaminobenzidine tetrahydrochloride (DAB) with 20 µg of catalase. Further method details are included in Appendix 1. Counts of COX-positive (COX^Pos), COX^Int, and COX^Neg myofibers were performed for the whole-muscle cross-section. COX-negative fibers are indicative of cells with high levels of mtDNA mutations and will thus not demonstrate the brown reaction product (oxidized DAB) during the first incubation but will stain blue following the second incubation for SDH activity. This is because the nuclear DNA entirely encodes SDH, so any mtDNA mutations will not affect its activity. In contrast, mtDNA mutations could affect complex IV activity and prevent DAB oxidation if a mutation affects a region of mtDNA containing the COX subunit genes. Similarly, COX^Int fibers exhibit low COX activity relative to SDH and appear bluish-gray, and are thought to represent muscle fibers/segments that are in the process of transition to COX^Neg status (*Murphy et al., 2012*).

## Mitochondrial DNA copy number

Groups of 25 fibers (5 × 5 fibers) in an unstained 20-µm-thick muscle cross-section from each subject were randomly selected (random number generator and numbered grid), laser captured, and their DNA extracted using the lysis method and stored at –20°C. The products were then separated and the bands visualized using a G-Box chem imaging system (*Figure 3—figure supplement 6—source data 1*A). The mtDNA fragment was extracted and the total mtDNA copy number in muscle fibers was determined using a standard curve (*Greaves et al., 2010*; *Figure 3—figure supplement 6—source data 1*B). The method details are in Appendix 1.

## Western blotting for mitochondrial proteins

Western blotting for representative mitochondrial proteins was performed as described previously (*Spendiff et al., 2016*). Briefly, 10–20 mg of muscle was homogenized in a Retch mixer mill (MM400) with 10× (w/v) of extraction buffer (50 mM Tris base, 150 mM NaCl, 1% Triton X-100, 0.5% sodium deoxycholate, 0.1% sodium dodecyl sulfate), and 10 µl/ml of Protease Inhibitor Cocktail. Following 2 hr of gentle agitation at 4°C, samples were centrifuged at 12,000 × *g* for 20 min at 4°C, and the supernatant removed for protein assessment by Bradford assay. Samples were diluted in 4× Laemmli buffer to yield a final protein concentration of 2 µg/ml and then boiled for 5 min at 95°C. Immuno-blotting was done using 20 µg of protein, loaded onto a 12% acrylamide gel, electrophoresed by SDS-PAGE and then transferred to polyvinylidene fluoride membranes (Life Sciences), blocked for 1 hr at room temperature in 5% (w/v) semi-skinned milk, and probed overnight at 4°C with the following primary antibodies (diluted in 5% BSA): mouse monoclonal anti-VDAC (1:1000; Abcam ab14734) and mouse monoclonal Total OXPHOS Cocktail (1:2000, Abcam ab110413). To address the poorer sensitivity to the CIV subunit in this cocktail after boiling human samples, we also probed using mouse monoclonal CIV (1:1000, Life Technologies A21348). Ponceau staining was performed to normalize protein loading. Following washing, membranes were incubated with HRP-conjugated secondary antibodies (diluted in 5% milk, Abcam) for 1 hr at room temperature. Protein bands were detected using SuperSignal West Pico Chemiluminescent Substrate (Thermo Scientific, USA) and imaged with a G-Box Chem Imaging System. Analysis of protein bands was performed using GeneTools software (Syngenem, UK).

## Statistical analyses

Statistical comparisons of physical function tests, vastus lateralis CSA, mtDNA copy number, and protein abundance by Western blot (VDAC) were performed using a two-tailed Student's *t*-test, with the p-value for significance set at <0.05. Statistical comparison of fiber type proportion (type × group), fiber size by type (type-specific size × group), Western blot (OXPHOS complex subunit abundance × group), and the abundance of respiratory chain compromised fibers (COX status × group) was performed by two-way ANOVA, with a Sidak multiple-comparison post-hoc test.

For LC-MS analyses, protein significance was determined with p-values derived from one-way ANOVA test to check any possible statistically significant difference between groups. The p-value threshold for a protein was considered as significant if p<0.05. Partial Least Square (PLS) analysis was used to derive models with the classification that maximized the variance between MA and NA groups. PLS loadings were derived from log2 normalized protein reporter ion intensity from all proteins. The statistical method was performed using R 3.3.6 with inbuilt libraries. Heatmaps and hierarchical cluster analyses were performed using the nonlinear minimization package in R. GraphPad Prism 6.07, and R Bioconductor packages were used for statistical analysis and generation of figures. STRING analysis (*Szklarczyk et al., 2019*) was used for obtaining protein-protein interaction network. Enrichment analysis was performed using ClueGO (*Bindea et al., 2009*) and PANTHER; the pathways were mapped and visualized using Cytoscape 3.7.2. One-way ANOVA, nonparametric, and chi-square tests (continuous and categorical variables) were used to test for sample differences.

**Table 1.** Characteristics of non-athletes (NA) and master athletes (MA).

| | NA (n = 14) | MA (n = 15) | p-Value |
|---|---|---|---|
| Age (years) | 80.9 ± 4.5 | 80.1 ± 4.8 | |
| Sex | | | |
| Male | 7 | 7 | |
| Female | 8 | 8 | |
| Body mass (kg) | 72.1 ± 11.4 | 62.2 ± 10.7 | 0.04 |
| Body fat (%) | 36.0 ± 6.6 | 21.9 ± 5.0 | <0.00 |

Values are mean ± SD.

**Table 2.** Training and competition history of octogenarian master athletes (MA).

| | n | Age (years) | Training per week (hr) | Years competing |
|---|---|---|---|---|
| Sprint, power | 8 (4F) | 79.9 ± 6.1 | 16 ± 3 | 16.6 ± 6.2 |
| Endurance | 7 (4F) | 80.3 ± 3.4 | 14 ± 3 | 26.6 ± 9.4 |

Values are mean ± SD.

## Results

### Superior clinical function in master athletes (MA) versus non-athlete controls (NA)

The general characteristics of the 15 MA and 14 NA participants are summarized in *Table 1*.

The athletes could generally be subdivided into two groups based upon their preferred competition events. Sprint/power athletes comprised individuals who competed in multisport jumping, throwing, and sprinting events, and individuals who competed in sprint running. Endurance athletes competed in track running and road running distances from 400 m to a full marathon (26.2 miles). An overview of the training and competition history of the MA group is given in *Table 2*. With respect to their training habits, it should be noted that each subject commented that the training load (particularly intensity) varied not only within a competition season but also within a 5-year age bracket (e.g., 75–79 years, 80–84 years). Training typically increased in the months approaching a birthday that would move them up to the next age category to take advantage of being the 'youngest' in their new age bracket at international competitions. In addition, regardless of the preferred competition events, all athletes noted a very mixed training regimen consisting of varying amounts of running, cycling, walking, stretching, yoga, and strength training. The rationale for selecting athletes from a broad array of athletics disciplines was that we were not interested in the effects of a specific type of exercise training per se (e.g., endurance or strength training), but rather in identifying individuals with exceptional physical capabilities regardless of their training. Consistent with this rationale, MA participants had superior function during the assessment of $VO_{2max}$, peak cycle work rate, time to walk 4 m fast, chair stands, timed-up and go, and balance time versus NA (*Figure 1a–f*), confirming that they represent high-functioning octogenarians.

### Greater preservation of muscle mass in octogenarian MA

All MA and NA participants underwent an MRI scan of the mid-thigh region at the same level as the muscle biopsy. Thigh cross-sectional images (*Figure 1g and h*) and MRI cross-sectional images of participants were analyzed (*Figure 1i*). The area of the vastus lateralis muscle (biopsied muscle) was determined for both legs. The estimated CSA of the vastus lateralis (average of both legs) was 30% higher in MA than NA (*Figure 1j*). Maximal isokinetic strength during knee extension was significantly greater in MA than NA. To consider the myosin genes that encode muscle mass maintenance and skeletal muscle contraction, we performed a fiber type proportion and fiber size type analysis (type I, type IIa, type IIx, and hybrid) by immunolabeling for the major MHC isoforms in MA and NA. This analysis shows no difference in fiber type proportion and a 28% higher mean fiber CSA in MA versus NA (*Figure 1—figure supplement 1a*). The lack of fiber type proportion differences between groups is corroborated by our proteomics data, which also shows no significant differences in the expression of MYH7 (type 1), MYH2 (type IIa), MYH1 (type 2x), and negligible expression of MYH4 (type IIb) as expected (*Figure 1—figure supplement 1b*). Indeed, after accounting for the false discovery rate (FDR), there were no significant differences in MHCs between groups. Furthermore, there were no significant differences in fiber size by type or in the type I to type II MHC protein expression ratio between MA and NA (*Figure 1—figure supplement 1c and d*, respectively).

### Quantitative proteomics reveals temporal proteome differences between MA and NA

To understand how skeletal muscle protein composition differs between MA and NA octogenarians, we performed a discovery proteomic analysis of muscle biopsies using LC-MS. We used a 10-plex

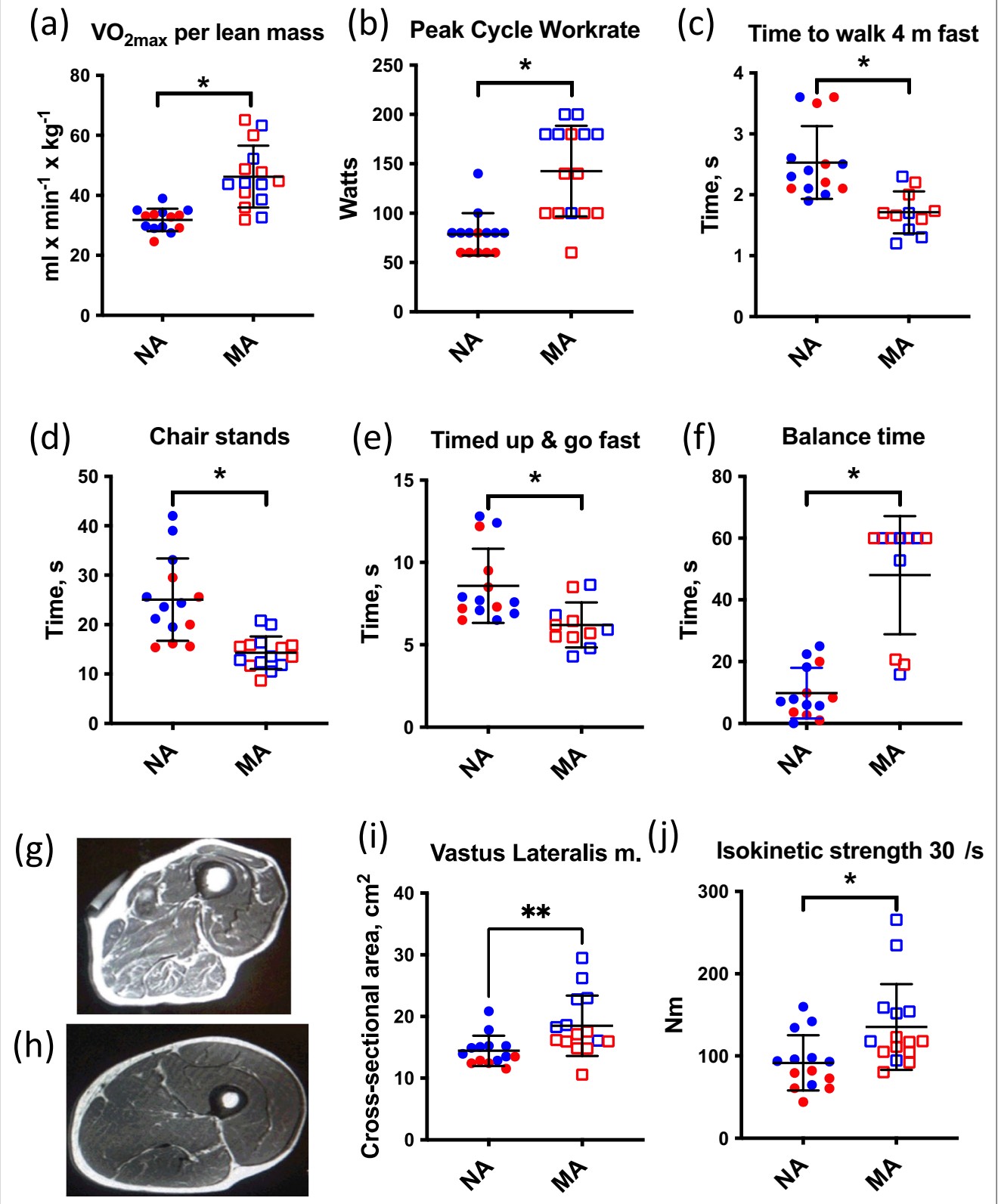

**Figure 1.** Muscle characteristics of master athletes (MA) and non-athletes (NA). (**a–f**) Clinical function tests in NA and MA. (**g**) Thigh cross-sectional image of an 80-year-old male NA (**h**) and an 83-year-old male MA. (**i**) Vastus lateralis muscle cross-sectional area (CSA) was greater in MA than NA. (**j**) Maximal isokinetic strength during knee extension was greater in MA than NA. Graphs show means and standard deviations. Groups were compared by a two-tailed Student's *t*-test, with <0.05.

*Figure 1 continued on next page*

*Figure 1 continued*

The online version of this article includes the following figure supplement(s) for figure 1:

**Figure supplement 1.** Fiber type and fiber size quantification.

TMT labeling approach that allows quantification and direct comparison between samples. Analyzing 28 participants, we were able to quantify 6176 proteins (*Figure 2a*, *Figure 2—figure supplement 1*). Of these, 4178 proteins (68%) were quantifiable across three TMT batches (present in all donors) and 1998 proteins (18%) were quantifiable in only one TMT batch (present in at least 10 donors). The quantitative protein expression between the TMT batches (*Figure 2b*) was mostly similar. The list of all proteins quantified from the MA and NA skeletal muscle is reported in *Supplementary file 1*. The partial least square (PLS) dimensionality reduction method used to stratify proteome distribution between MA and NA from 24 donors (*Figure 2c*) reveals a clear separation between the groups along the PC1 (11.6%) and PC2 (16.7%) axes and PC3 (11.1%) axes.

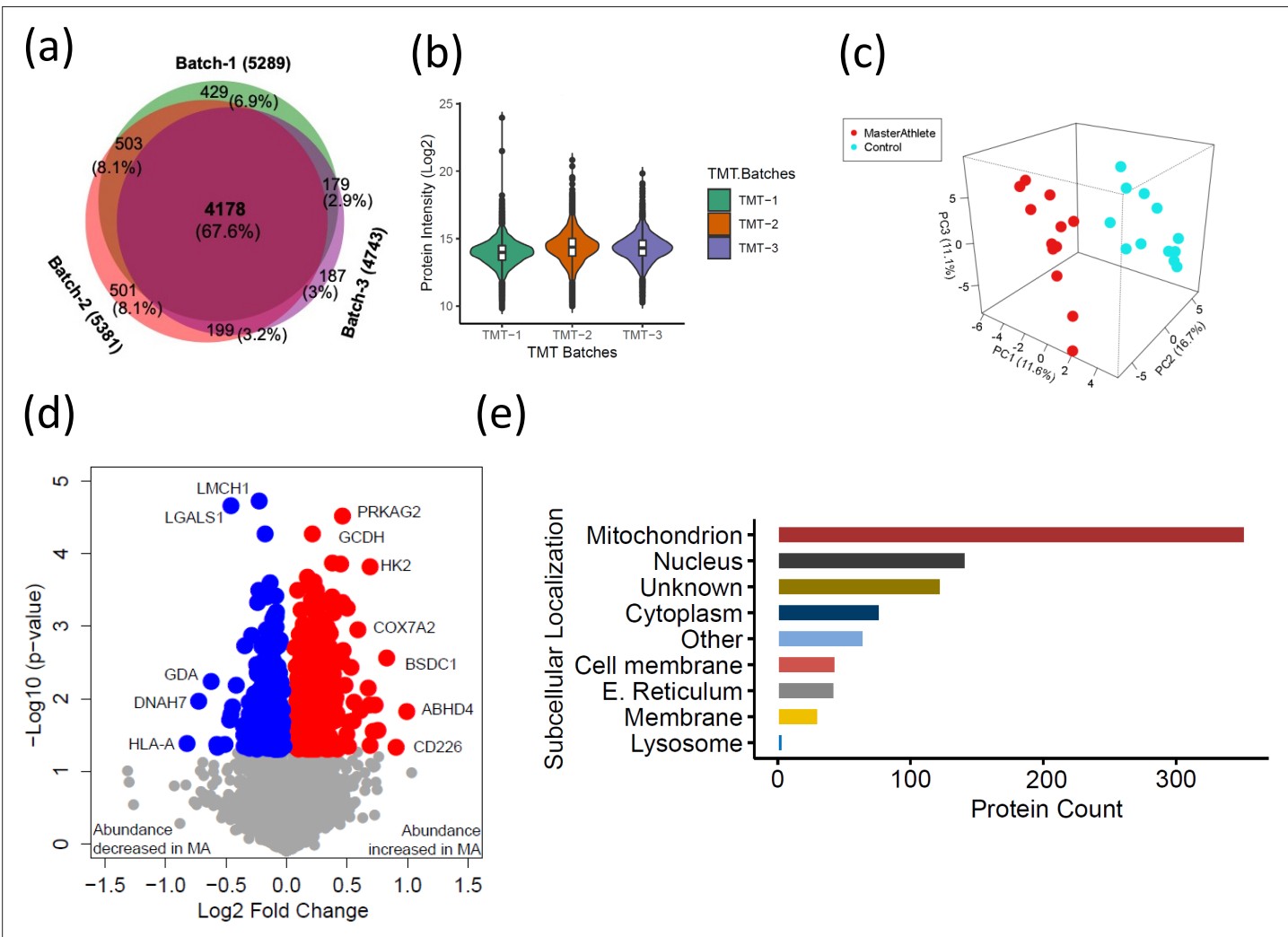

**Figure 2.** The quantitative proteome reveals temporal proteome changes between master athletes (MA) and non-athletes (NA). (**a**) Number of proteins quantified among three tandem mass tag (TMT) batches. (**b**) Quantitative protein expression between three TMT batches. (**c**) Partial least square (PLS) plot of MA and NA donors. Red circles are MA donors, and cyan circles are NA donors. (**d**) Proteins differentially expressed between MA and NA. Each circle is a protein, red circles are proteins increased in abundance in MA, and blue circles are proteins decreased in abundance in MA. (**e**) Cellular location of the differentially expressed proteins in MA and the number of proteins encoded for each component are shown (X-axis).

The online version of this article includes the following figure supplement(s) for figure 2:

**Figure supplement 1.** Normalized tandem mass tag (TMT) batches.

Of all the 6176 proteins quantified, 880 were differentially represented between MA and NA (Student's *t*-test, p<0.05, fold change [FC] > 1.02 for overrepresented proteins and <0.9 for under-represented proteins), and of these, 544 proteins were overrepresented and 336 proteins were under-represented in MA compared to NA (*Figure 2d*, *Supplementary file 2*). UniProt cellular localization coverage for these divergently represented proteins is shown in *Figure 2e*. Of note, 42% of the total 880 significantly altered proteins in octogenarians were mitochondrial proteome, and most of the differentially represented proteins relate to mitochondrial structure or OXPHOS. This ample coverage of the mitochondrial proteome enables us to explore the modulating role of mitochondria in high-functioning octogenarians' muscle metabolism.

## Mitochondrial protein enrichment in octogenarian MA

The 369 mitochondrial proteins overrepresented in MA include 117 mitochondrion inner membrane proteins, 21 outer membrane proteins, 18 matrix proteins, 10 inter-membrane space proteins, and 5 outer membrane proteins. The abundance of all mitochondrion proteins is higher in MA, except eight proteins (*Figure 3a*). Enrichment analysis with the whole human genome as a statistical back-ground revealed oxidoreductase activity, electron transport activity, and cofactor binding as the top significantly enriched pathways in MA after FDR correction and Fisher's exact test cutoff at p<0.01 (*Figure 3b*). Specifically, 110 proteins associated with TCA and respiratory electron transport, 71 proteins from OXPHOS and 43 protein constituents of complex I, 3 in complex II (SDHA, SDHB, SDHC), 8 in complex III, 13 in complex IV, and 10 in complex V were significantly more abundant in MA (*Figure 3c*).

The cytoplasmic and nuclear SIRTs were not quantified in our dataset; however, we explored SIRT3 and SIRT5 mitochondrial sirtuins, which are master regulators of mitochondrial biology, including ATP production, metabolism, apoptosis, and intracellular signaling. Both SIRT3 and SIRT5 proteins were 1.2-fold more abundant in MA than NA (p<0.01) (*Figure 3d*). Of note, the overrepresentations of SIRT3 in MA were consistent with higher deacetylation of long-chain acyl-CoA dehydrogenase (LCAD) in MA (FC 1.14 and p=0.007), which suggest elevation of lipid catabolism and fatty acid oxidation pathways. The deacetylase activity of SIRT3 improves mitochondrial function by the deacetylation of mitochondrial complex I protein NADH ubiquinone oxidoreductase subunit A9 (NDUFA9) (*Ahn et al., 2008*) and succinate dehydrogenase from complex II (SDH) (*Cimen et al., 2010*). SIRT3 also deacetylates the mitochondrial permeability transition-regulating protein, cyclophilin D, to reduce likelihood of opening of the mitochondrial permeability transition pore (*Hafner et al., 2010*). Finally, SIRT3 deacetylates lysine residues on SOD2 to promote its antioxidant activity and thereby reduce the level of reactive oxygen species (ROS) released outside mitochondria. While we would expect this deacetylation to increase SOD2 activity independent of changes in SOD2 content, in our study SOD2 protein (FC = 1.17, p=0.037) was also more highly expressed in MA. Comparatively less is known about SIRT5 than SIRT3, but it has been reported that SIRT5 physically interacts with cytochrome *c* (CYCS) and CYCS abundance was 1.3-fold higher in MA (*Figure 3—figure supplement 1a*).

While our proteomics analyses identified a globally higher abundance of OXPHOS proteins (*Figure 3c*), markers of mitochondrial content specifically assessed by Western blot were not univo-cally associated with MA status. For example, VDAC was not different between groups, whereas citrate synthase by proteomics was elevated in MA (*Figure 3—figure supplement 1b*). Further-more, we observed a significant effect (p=0.046) for a higher abundance of OXPHOS complexes relative to VDAC in MA when analyzed by Western blot, consistent with the higher abundance of OXPHOS complexes by proteomics in MA (*Figure 3—figure supplement 1c*; uncut blots for VDAC and OXPHOS subunits are shown in *Figure 3—figure supplement 2—source data 1*). Histochem-ical analysis was performed to quantify muscle fibers with compromised respiratory function based upon the ratio of staining intensity of COX (contains three mtDNA-encoded subunits and thus sensi-tive to a high burden of mtDNA mutations) relative to SDH (entirely nDNA-encoded). Specifically, muscle fibers with low COX relative to SDH (COX^Int) and deficient COX relative to SDH (COX^Neg) were considered to have compromised respiratory function. This analysis revealed a significantly higher abundance of healthy COX^Pos fibers (p=0.0291) and fewer respiratory chain compromised (COX^Int) myofibers (p=0.0448) in MA (*Figure 3e and f*). Thus, the proteomics data is consistent with histochem-ical phenotypic data showing better maintenance of respiratory competent muscle fibers (COX^Pos fibers) in MA and a greater abundance of ETC subunits relative to VDAC. This latter observation could

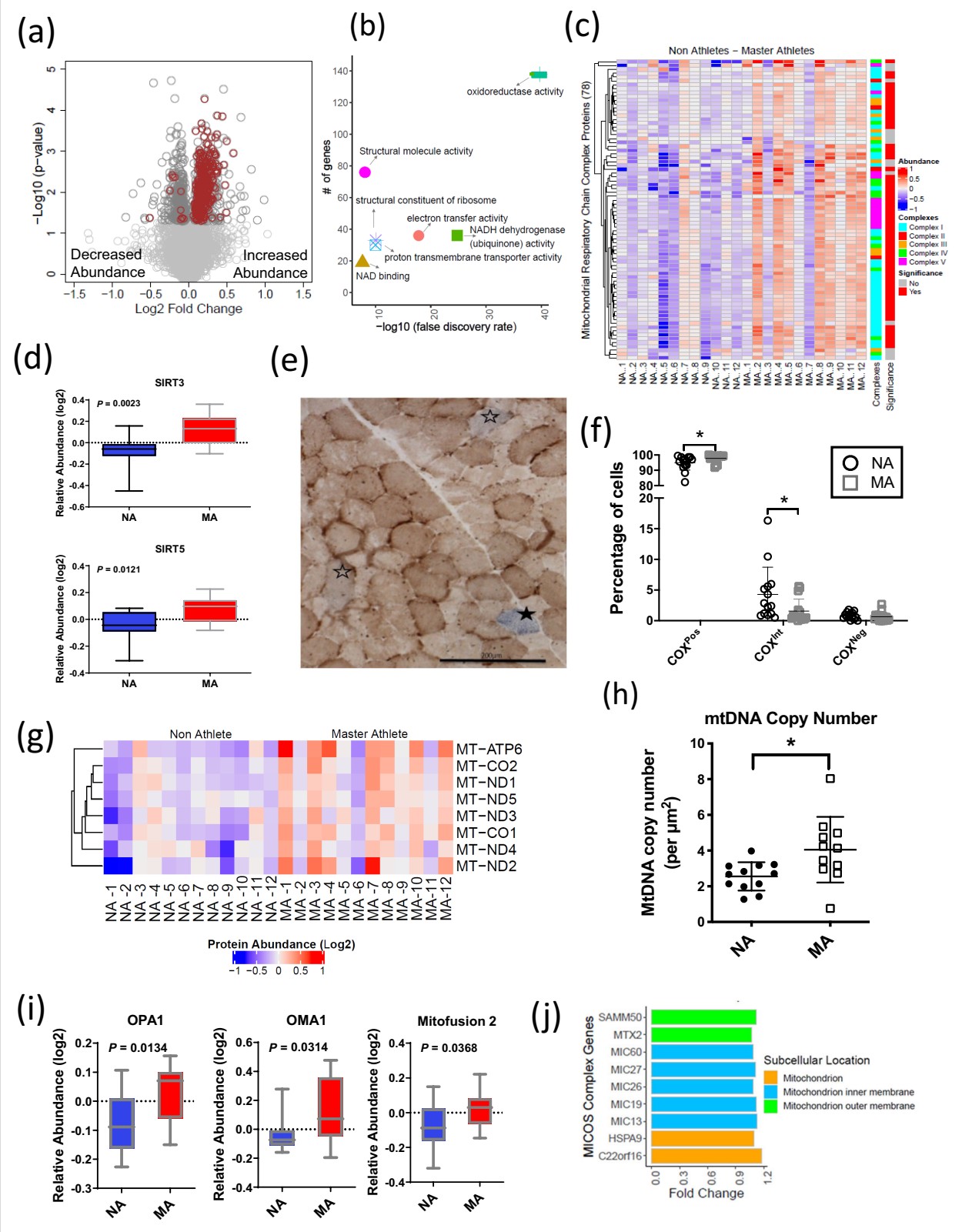

**Figure 3.** Mitochondrial protein enrichment in octogenarian master athletes (MA). (**a**) Dysregulation of significant mitochondrial proteins shown as red circles. (**b**) Functional classification of mitochondrial proteins with protein-protein interaction enrichment p-value<1.0e-16. (**c**) Heatmap showing upregulated respiratory chain complex proteins in MA. 71 complex proteins on y-axis. X-axis shows donors. (**d**) Enrichment of mitochondrial sirtuins SIRT5 and SIRT3 in muscle of MA versus non-athletes (NA). (**e**) Respiratory chain compromised fibers in skeletal muscle. COX/SDH image showing the

*Figure 3 continued on next page*

identification of COXPos (brown cells), COXInt (empty stars), and COXNeg muscle fibers (solid star). COXNeg fibers have lost complex IV activity relative to SDH and appear blue, COXInt retain small amounts of COX activity relative to SDH and appear gray, and COXPos fibers have normal COX activity relative to SDH and appear brown. Scale = 200 µm. (**f**) Quantification revealed a significantly higher abundance of healthy COXPos fibers (*p=0.0291) and fewer respiratory chain compromised (COXInt) myofibers (*p=0.0448) in MA compared to NA. (**g**) Upregulation of mitochondrial DNA (mtDNA) in MA. MA and NA donors are shown on X-axis; quantified mtDNA proteins are shown on Y-axis. (**h**) Increased mtDNA copy number in MAs. Absolute mtDNA copy number was determined using a standard curve constructed from known amounts of mtDNA. MA had significantly more copies of mtDNA than NA (*p=0.0177; *t*-test). Graph shows the means and standard deviation. (**i**) Protein groups that maintain the functional integrity of mitochondria were higher in MA. (**j**) Upregulated MA proteins in MICOS complex system and the fold change of the proteins. Cellular location of the proteins is color coded.

The online version of this article includes the following source data and figure supplement(s) for figure 3:

**Figure supplement 1.** Abundance of mitochondrial proteins, VDAC, and subunits of the oxidative phosphorylation (OXPHOS) chain assessed by Western blot and mass spectrometry (MS) in master athletes (MA) versus non-athletes (NA).

**Figure supplement 2.** Mitochondrial protein quantification.

**Figure supplement 2—source data 1.** Uncut blots for VDAC and oxidative phosphorylation (OXPHOS) subunits.

**Figure supplement 2—source data 2.** Source data for the Western blotting of oxidative phosphorylation (OXPHOS) subunit and VDAC proteins are found in *Figure 3—figure supplement 2*.

**Figure supplement 3.** Mitochondrial DNA (mtDNA) enrichment analysis and cristae formation.

**Figure supplement 4.** Autophagy lysosomal system and ubiquitin proteasome pathway proteins.

**Figure supplement 5.** Nuclear pore membrane proteins.

**Figure supplement 6.** Generating a standard curve in order to determine absolute mitochondrial DNA (mtDNA) copy number.

**Figure supplement 6—source data 1.** Mitochondrial DNA copy number determination blots.

suggest greater cristae surface area relative to mitochondrial volume or differences in the clearance of mitochondrial membranes.

In contrast to the general higher abundance of mitochondrial proteins noted above, eight mitochondrial proteins had a lower abundance in MA, which were NADH-cytochrome b5 reductase 3 (CYB5R3), phosphatidate cytidylyltransferase 2 (CDS2), long-chain-fatty-acid--CoA ligase 3 (ACSL3), dimethylarginine dimethylaminohydrolase 1 (DDAH1), WD repeat-containing protein 26 (WDR26), serine/threonine-protein phosphatase PGAM5 (PGAM5), SHC-transforming protein 1 (SHC1), and StAR-related lipid transfer protein 7 (STARD7).

## mtDNA protein enrichment and maintenance of cristae architecture in octogenarian MA

Previous studies suggest that respiratory chain defects in skeletal muscle may result from high levels of mtDNA mutations (*Bua et al., 2006*; *Murphy et al., 2012*) and/or mtDNA depletion (*Müller-Höcker et al., 1993*; *Mueller et al., 2012*). To address this issue in our subjects, we specifically explored mitochondrial proteins in our proteomics dataset encoded in mtDNA. Of the known 13 mtDNA proteins, 8 were quantified in our data, and all of them were significantly more abundant in MA than in NA (p<0.05) (*Figure 3g*). The proteomics data were consistent with findings that absolute mtDNA copy number evaluated using a quantitative method was higher in MA than in NA (*Figure 3h*) and indicated parallel greater abundance of mtDNA copies and mtDNA-encoded proteins in MA. Further, the observation of a lower abundance of respiratory compromised fibers (defined as low or absent complex IV staining in COX-SDH double-stained muscle cross-sections) (*Figure 3e and f*) in MA versus NA is consistent with a lower burden of mtDNA mutation in highly functioning MA octogenarians compared to NA.

Consistent with the higher protein levels of many mitochondrial proteins in MA, our results show that 38 proteins from 28S and 39S mitoribosomal proteins were significantly more abundant in MA, suggesting an increased mitochondrial protein synthesis. Conversely, cytoplasmic ribosomal protein (RPS2, RPLP0) abundance was lower in MA, suggesting reduced cytoplasmic ribosome protein synthesis (*Figure 3—figure supplement 1d*).

Mitochondrial morphology is regulated by proteins that modulate fission (e.g., DRP1) and fusion (e.g., OPA1, MFN1 and 2). For example, OPA1 induces mitochondrial inner membrane fusion (*Mishra et al., 2014*) to promote cristae tightness, increase the activity of respiratory enzymes, and enhance

the efficiency of mitochondrial respiration (*Cogliati et al., 2013*). Interestingly, OPA1, MFN1, and DRP1 were overrepresented in MA (*Figure 3i*), although DRP1 fold elevation in MA donors was not statistically significant. The mitochondrial contact site and cristae organizing system (MICOS) complex are crucial for maintaining cristae architecture, and experimental knockdown of MICOS components leads to mitochondria with altered cristae morphology and compromised OXPHOS (*Friedman et al., 2015*; *Yang et al., 2015*). In this study, 15 out of 17 UniProt annotated MICOS complex proteins were quantified, and 9 of them were significantly more abundant in MA (*Figure 3j*). For example, the mitochondrial inner membrane protein mitofilin (MIC60), which controls cristae morphology and is thus indispensable for normal mitochondrial function (*John et al., 2005*), was 1.2-times-fold more abundant in MA donors. Of note, we have previously reported a decrease in the abundance of these nine proteins with healthy aging (*Ubaida-Mohien et al., 2019b*).

A complex array of dynamic protein interactions (Sam50, Metaxin, and the inner membrane-localized MICOS) at cristae junctions that form the mitochondrial intermembrane space bridging (MIB) complex was reported recently (*Huynen et al., 2016*). The outer mitochondrial membrane protein Metaxin2 (MTX2), which was significantly more abundant in MA (*Figure 3—figure supplement 3a*), interacts with MICOS complex and MTX3, which are the part of MIB complex (*Huynen et al., 2016*). Metaxins, together with Sam50, are also important for the stability of respiratory complexes (*Ott et al., 2012*). A general translocase mediates the import of nuclear-encoded mitochondrial prepro-teins in the outer membrane, the TOM complex, and by two distinct translocases in the mitochondrial inner membrane, the TIM23 complex, and the TIM22 complex. The average expression of 2 TOM complex proteins (TOMM22 and TOMM40) and 10 TIM complex proteins (TIM10, TIM13, TIM14, TIM16, TIM21, TIM22, TIM23, TIM29, TIM44, and TIM50) was found to be more abundant in MA (*Figure 3—figure supplement 3b*).

## Autophagy and proteostasis pathway proteins in octogenarian MA

Skeletal muscle mass is influenced by the proteolytic process of protein turnover and degradation. The major regulatory process of the proteolytic system is chaperone-mediated autophagy by lyso-somes and the ubiquitin proteasome pathway. There were 267 proteins from these pathways quan-tified and 47 proteins were significantly associated with MA ($p < 0.05$, 17 underrepresented in MA). The proteins were categorized as autophagy, autophagy-lysosome, chaperones, proteasome, and other proteostasis cluster proteins (*Figure 3—figure supplement 4*). Proteasome proteins PSMB1, PSMA2, small heat shock protein HSPB8, DNAJ proteins like DNAJB4 and DNAJC3, were lower in MA. Activation/inhibition of autophagy – such as V-type proton ATPase 116 kDa subunit isoform 1 (ATP6V0A1), heat shock 70 proteins like HSPA2 and HSPA1A proteins – were also lower in MA. A lower ATP6V0A1 was reported previously in highly active aging healthy donors (*Ubaida-Mohien et al., 2019a*). In contrast, many mitochondrion-localized proteostasis proteins like HSCB, MRPL18, TIMM9, HSPE1, and HSPA9 were higher in abundance in MA. PRKAG2, 5′-AMP-activated protein kinase subunit gamma-2, a component of AMP kinase main energy-sensor protein kinase that responds to changes in the cellular AMP:ATP ratio and regulates the balance between ATP produc-tion and consumption, was one of the highly expressed proteins (log2FC 1.3) in MA octogenarians, suggesting a tightly monitored balance between energy production and utilization (*Mounier et al., 2015*).

## Impact of nuclear pore membrane proteins and transport proteins in octogenarian MA

Nuclear pore complexes (NPCs) facilitate and regulate the transport of different macromolecules across the nuclear envelope, allowing bilateral exchanges between the nuclear and cytoplasmic environment (*Strambio-De-Castillia et al., 2010*; *Wente and Rout, 2010*). 25 nuclear pore proteins were quantified, all less expressed in MA than in NA, and for 12 of them, the difference was statis-tically significant ($p < 0.05$) (*Figure 3—figure supplement 5a*). Nucleopore cytoplasmic filaments like NUP358, NUP98, and NUP88, and adaptor NUPs like NUP98/96 were less abundant in MA. Tpr, the central architectural element of nuclear pore formation, Nup93, which is critical for nuclear perme-ability, was also less abundant in MA (*Figure 3—figure supplement 5b*). The lower abundance of proteins of the nuclear pore in MA was unexpected and should be further explored in future studies.

## Spliceosome pathway proteins are underrepresented in octogenarian MA

Alternative splicing produces protein variants by combining information from different exon sequences in the same genes. Aging is associated with the emergence of different splicing variants of the same genes (*Harries et al., 2011*; *Holly et al., 2013*; *Bhadra et al., 2020*). However, it remains unknown whether these changes in the human proteome are part of the aging process or represent resilience strategies to cope with the damage accumulation and functional decline associated with aging (*Deschênes and Chabot, 2017*). Previous studies have shown that alternative splicing is particularly abundant in skeletal muscle, and we have shown that proteins that regulate alternative splicing are significantly overrepresented in skeletal muscle tissue from older compared to younger healthy individuals (*Ubaida-Mohien et al., 2019b*). Interestingly, after accounting for age and other covariates, being physically active in daily life was associated with a lower representation of spliceosome proteins

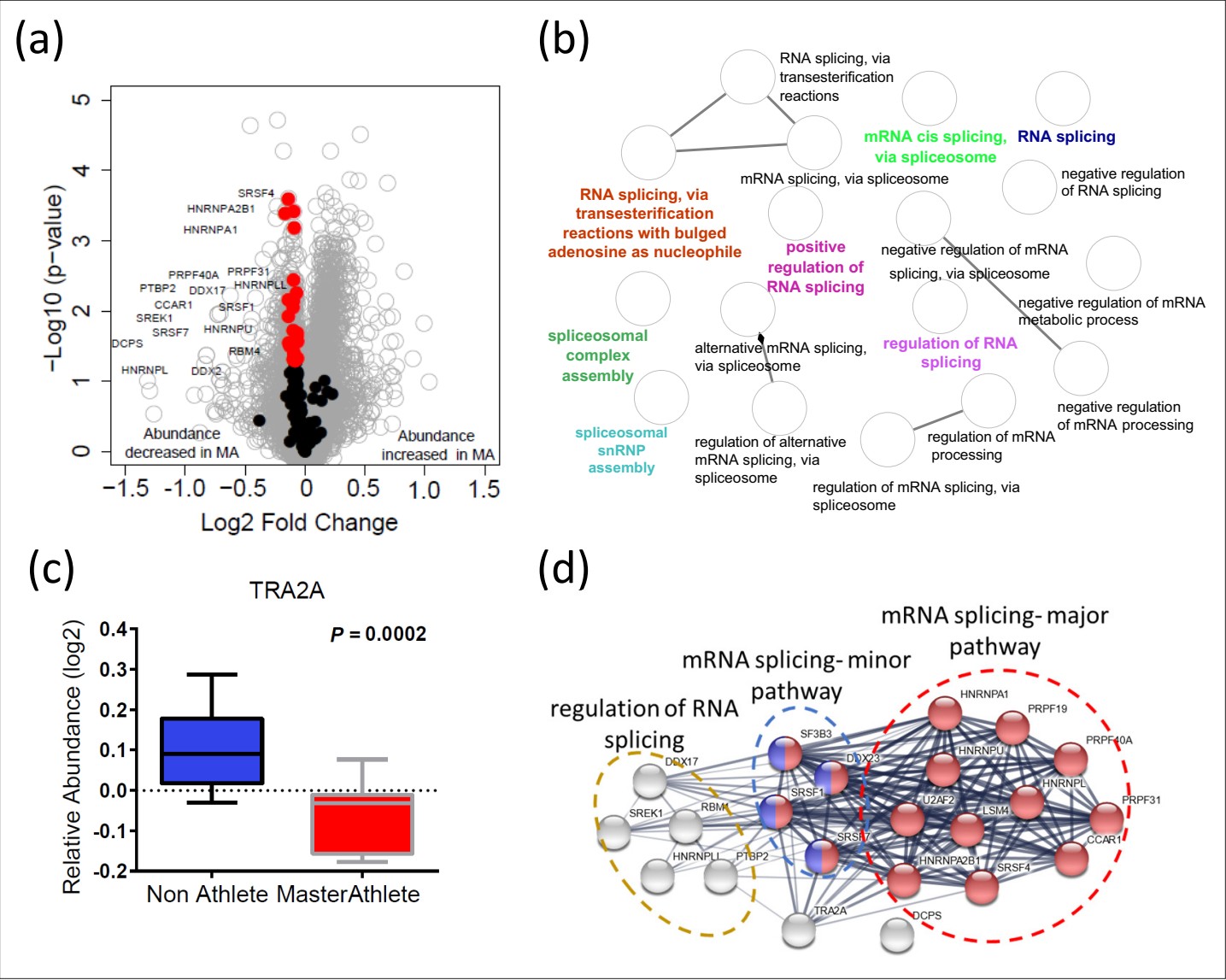

**Figure 4.** Dysregulation of spliceosome pathway proteins in octogenarian master athletes (MA). (**a**) Underrepresentation of spliceosome pathway proteins. Significant spliceosome proteins (22) underrepresented in MA are marked as red circles, and all other proteins are marked as gray circle. X-axis shows log2 fold expression of the proteins in MA versus non-athletes (NA). (**b**) The functional characteristics of the spliceosome proteins are shown. Each GO annotation cluster is color coded. (**c**) Downregulation of TRA2 protein in MA donors. Y-axis shows the log2 relative protein abundance. (**d**) Interaction partners of TRA2 protein; with RNA splicing regulation proteins, mRNA major splicing pathway, and mRNA minor splicing pathway proteins.

in skeletal muscle (*Ubaida-Mohien et al., 2019a*). Also, one of the strongest signals in the current analysis was a lower representation of proteins related to mRNA metabolic process, mRNA splicing, and mRNA processing in MA. In particular, we were able to quantify 132 spliceosome proteins, and of these 22 proteins were less abundant in MA (p<0.05) (*Figure 4a*). The functional characteristics of the spliceosomal proteins are shown in *Figure 4b*. Of note, TRA2A, an RNA-binding splicing factor protein that modulates splicing events and translation, was among the significantly affected proteins (p=0.0004) in this category and also had the greatest fold-difference from NA (*Figure 4c*). Functional analysis of TRA2A suggests a highly interconnected functional interaction network with two major pathway proteins: mRNA splicing major pathway (red) and mRNA splicing minor pathway (blue) proteins (*Figure 4d*). Despite not having a direct interaction within this network, the DCPS protein has a role in the first intron splicing of pre-mRNAs (*Figure 4d*). Taken together, the lower representation of spliceosome proteins that regulate alternative mRNA splicing in MA is consistent with the effects of physical activity in our previous study and is consistent with our previous hypothesis that alternative splicing is part of a resilience response in the face of lower mitochondria function (*Ferrucci et al., 2022*). Such a resilience response is not required in MA because of high mitochondrial function maintenance despite old age (*Ubaida-Mohien et al., 2019a*). This hypothesis is also consistent with previous data showing that after adjusting for age and physical activity better skeletal muscle oxidative capacity assessed by $^{31}$P-MR spectroscopy is associated with overrepresentation of splicing machinery and pre-RNA processing proteins (*Adelnia et al., 2020*).

## Modulation of mitochondrion and splicing machinery with aging, physical activity, and exercise

To further explore the hypothesis that alternative splicing is part of a compensatory adaptation to impaired mitochondrial function, we combined the results of this study with the skeletal muscle proteomic data in 58 healthy participants collected in the GESTALT study (*Figure 5—source data 1*; *Ubaida-Mohien et al., 2019a*; *Ubaida-Mohien et al., 2019b*). Notably, we searched for proteins that were underrepresented with age (GESTALT, Age-) and overrepresented with both higher physical activity (GESTALT, PA+) and in master athletes (MA+) compared to age-matched controls (*Figure 5*). Enrichment analysis of proteins at the intersection showed 50 proteins enriched at all three shared interceptions, including proteins representing mitochondrial biogenesis, TCA cycle and respiratory electron transport, MICOS complex, and cristae formation (*Figure 5*).

Although there was considerable overlap between proteins overrepresented with higher physical activity in the GESTALT study and proteins overrepresented in MA, a large group of proteins related to mitochondrial protein import and mitochondrion organization were specific to the MA group (not associated with physical activity per se). This suggests that although many of the proteins that are more abundant in MA versus NA can be attributed to MAs' physical activity habits, this does not account for all of the differences observed. Specifically, out of 301 unique MA+ proteins, a subset of proteins – mitochondrial translation (36 proteins), mitochondrial inner membrane (75 proteins), and mitochondrial matrix proteins (65 proteins) – appear unrelated to physical activity and may reflect unique biology in our MA group (*Figure 6*).

In the next analysis, we considered the proteins that were overrepresented with age (GESTALT, Age+) and underrepresented with both higher physical activity (GESTALT, PA-) and in master athletes (MA-) compared to age-matched controls (*Figure 7a*). Interestingly, we found 40 proteins in all three meaningful interceptions (Age+/PA-/MA-; Age+/PA-/; Age+MA-), and these involved mRNA splicing, capped introns containing pre-RNA, sarcolemma, regulation of glucokinase, spliceosome, and metabolism of RNA (*Figure 7*). The other notable category pathway differentially represented in Age+ and MA- was the NLRP inflammasome pathway, indicating the inflammasome proteins that increase with aging are antagonized in MA subjects. Although more proteins were affected in PA than MA versus NA, several proteins were underrepresented in MA versus NA that were not underrepresented in PA, supporting the idea that there are likely factors beyond physical activity involved in protecting the MA group's muscle proteome. Specifically, 162 MA-exclusive proteins were underrepresented in MA versus NA and reflect the unique physiology of MA participants. Enrichment analysis identified proteins regulating nuclear pore organization (NUP133, NUP153, NUP54), heterochromatin organization (HP1BP3, H3F3B, and HMGA1) and telomere (HMBOX1, PURA, TERF2IP), mRNA and splicing process, and contractile/sarcomere fiber proteins (*Figure 7b*, *Supplementary file 3*).

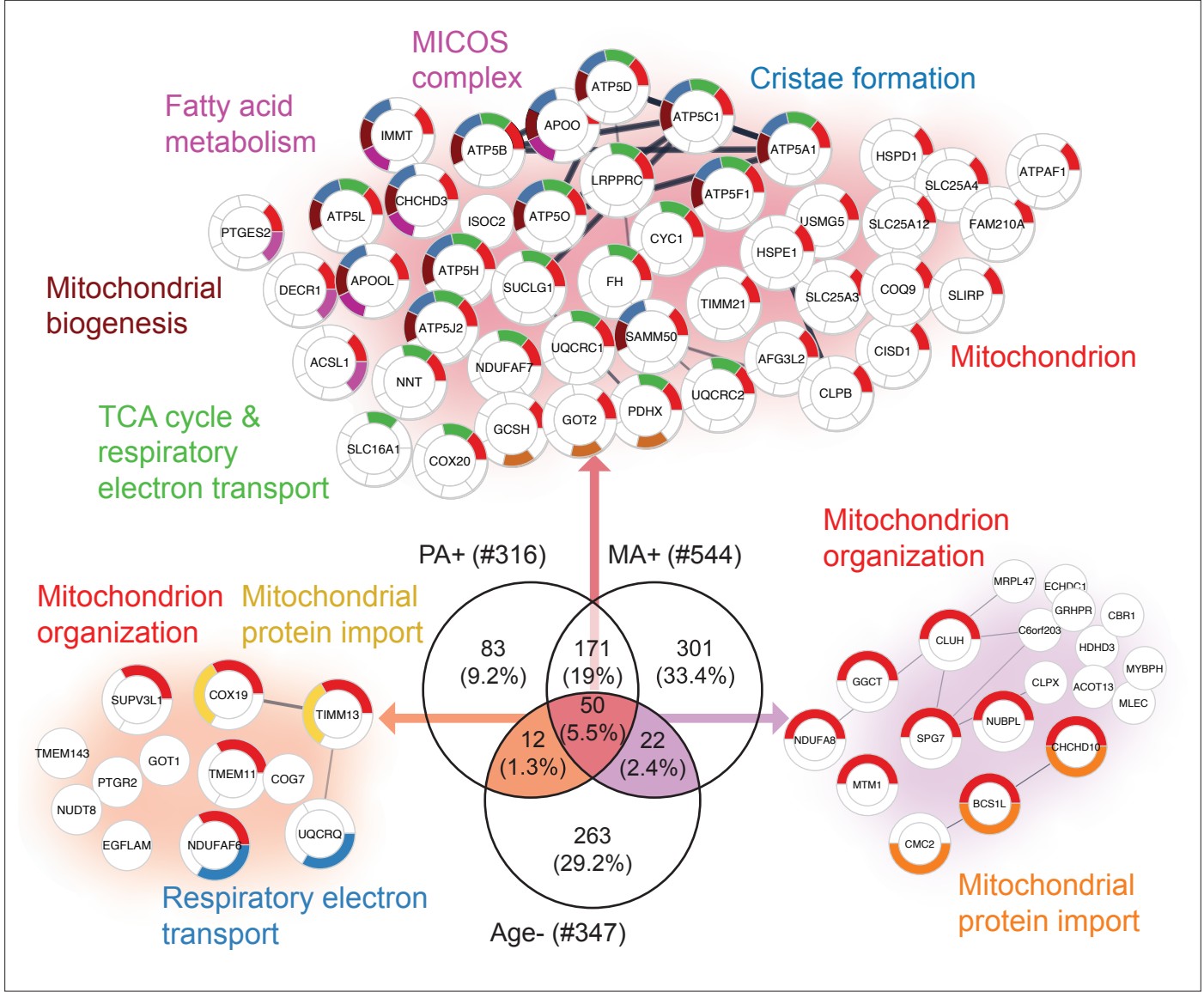

**Figure 5.** Aging proteins countered by physical activity (PA) and master athletes (MA). Proteins that decline with aging but are antagonized in physically active subjects (GESTALT, n = 58) and MA group. Enriched pathways from 50 proteins that increase with PA and MA and decrease with Age are shown (top), and pathways from 12 proteins are are in common between PA and Age (left) and enriched pathways from 22 proteins that are in common between MA and Age (right). Top enriched pathways are color coded (significance threshold false discovery rate [FDR] < 0.05). Proteins without interaction partners are omitted from visualization.

The online version of this article includes the following source data for figure 5:

**Source data 1.** Baseline characteristics of the GESTALT skeletal muscle participants.

# Discussion

MS-based proteomics studies strongly suggest in a select group of individuals free of major disease risk factors and morbidity that skeletal muscle mitochondrial proteins are underrepresented in older compared to younger persons, and, independent of age, are overrepresented in those who are more physically active in daily life (*Kleinert et al., 2018*; *Ubaida-Mohien et al., 2019a*; *Ubaida-Mohien et al., 2019b*). In this respect, these prior studies suggest that aging and physical activity have opposite effects on mitochondrial health. However, as most individuals' level of physical activity declines with aging, a clear-cut dissection of the effect of aging independent of declining physical activity has proven difficult to achieve. To address this question, 15 exceptionally fit and physically very active octogenarian MA were compared to 14 healthy but non-athletic octogenarian NA. In accordance with

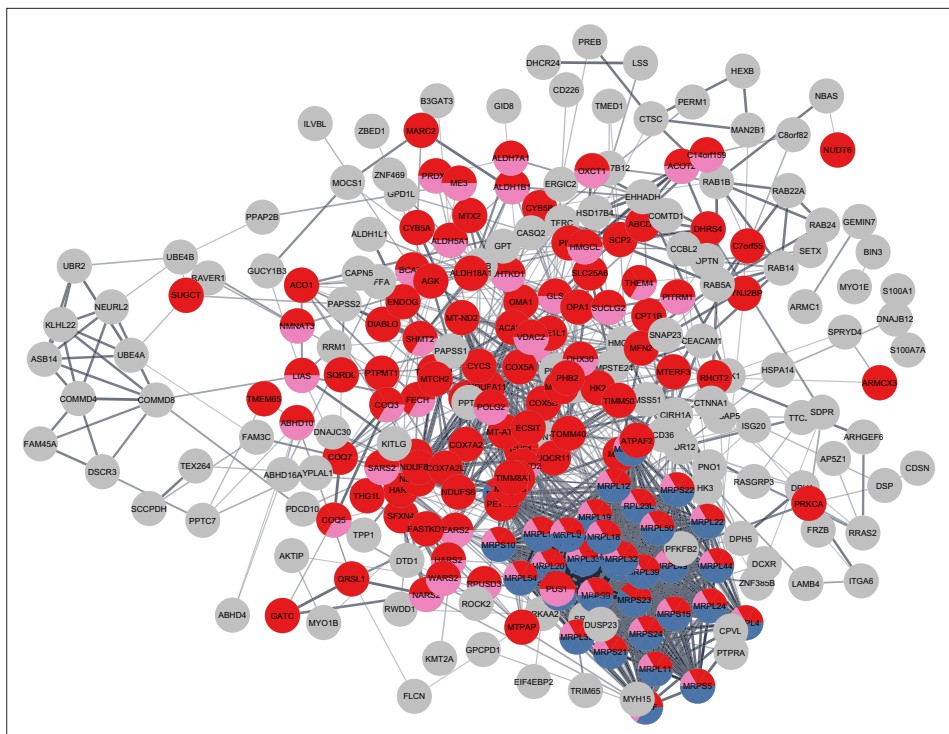

**Figure 6.** Master athletes (MA)-exclusive overrepresented proteins. The subset of 301 MA+ proteome represents clusters of mitochondrial translation pathway enrichment (blue circles, 36 proteins), mitochondrial inner membrane (red circles, 75 proteins), and mitochondrial matrix proteins (pink, 65 proteins). Mitochondrial translation pathway proteins are localized either as inner membrane proteins or as matrix proteins. Each circle node is a protein exclusive to MA from (MA+, PA+, and Age-) analysis, Nodes without any interaction are excluded from the enrichment analysis.

our hypothesis, we found an overrepresentation of mitochondrial proteins and these data were consistent with the finding of higher mtDNA copy number, fewer respiratory chain compromised muscle fibers by histochemistry, and an increased ratio of mitochondrial inner membrane-bound ETC subunits relative to the outer mitochondrial membrane protein VDAC in skeletal muscle of MA octogenarians. We also found a lower abundance of proteins regulating RNA splicing in MA, confirming that in older persons with high mitochondrial function the upregulation of the splicing machinery usually observed with older age is not occurring (*Ubaida-Mohien et al., 2019b*).

In general, we found that many proteins overrepresented in MA were similar to the proteins that have been associated with higher physical activity in daily life, independent of age in previous studies (*Ubaida-Mohien et al., 2019a*). These findings demonstrate that some of the biological mechanisms that facilitate the high function of our octogenarian MAs in spite of old age are similar to those beneficially affected by moderate physical activity in skeletal muscle (*Nilsson and Tarnopolsky, 2019*). However, we also found differentially represented proteins in highly functioning octogenarian MA that were unique from those affected by physical activity, suggesting that unique biological mechanisms also contribute to the extreme performance capacity in this select group of individuals. This unique set of proteins unrelated to physical activity may provide novel insight on mechanisms, either genetic and/or linked to life-course exposure, that may lessen the physical function decline that is observed in the great majority of aging individuals. In total, we found 176 proteins related to mitochondria that were overrepresented in MA versus NA that had not been previously linked to physical activity. For example, amongst these, 22 proteins that mediate mitochondrial protein import and are involved in establishing and modulating the mitochondrial architecture were overrepresented in MA but had not been previously reported as affected by physical activity (*Ubaida-Mohien et al., 2019a*).

A striking result of this study is that 80 proteins involved with mRNA splicing, metabolism of RNA, capped intron containing pre-RNA, and transcription coregulator activity that were shown previously to significantly increase with aging (*Rodríguez et al., 2016*; *Ubaida-Mohien et al., 2019b*)

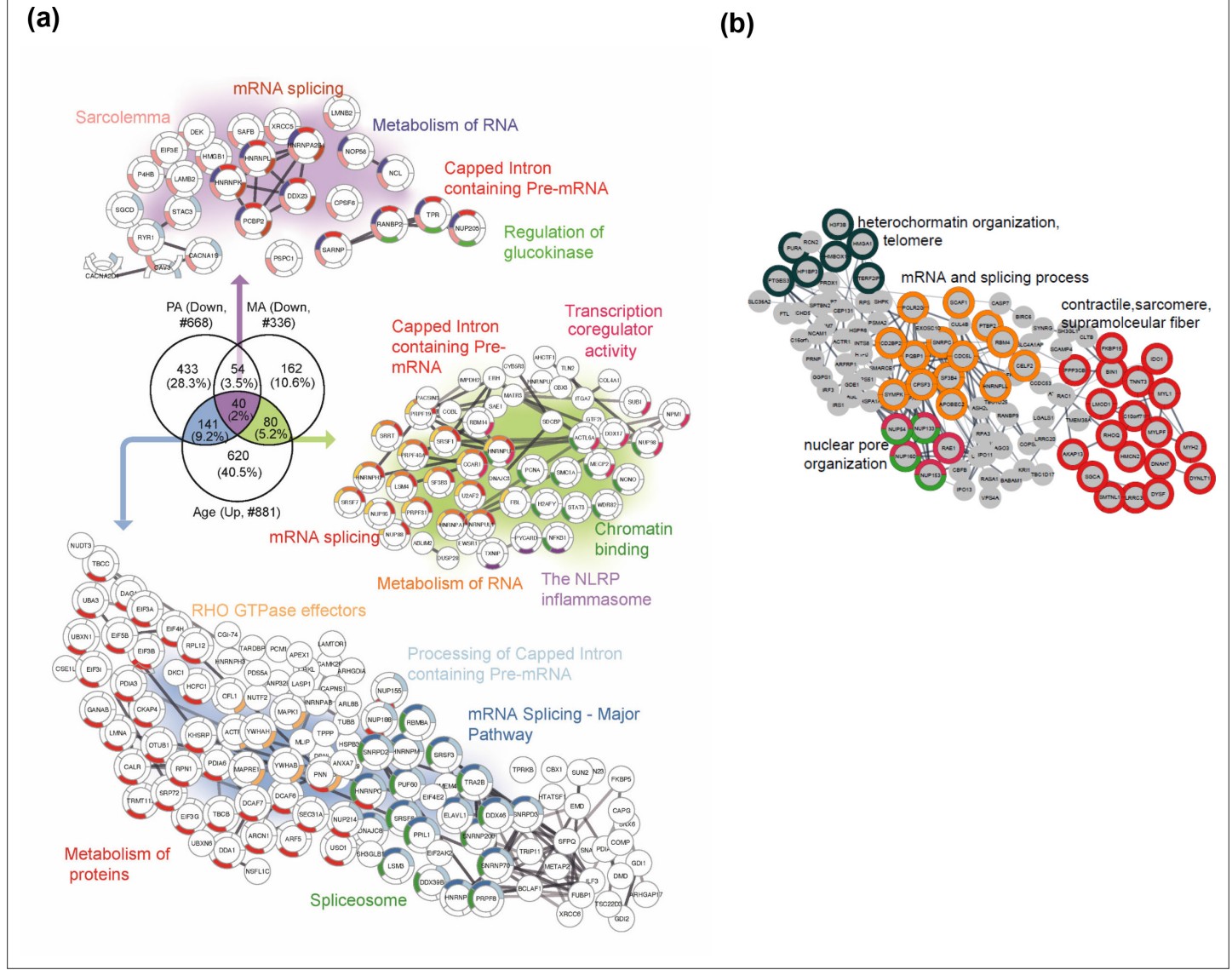

**Figure 7.** Master athletes (MA)-exclusive underrepresented proteins. (**a**) Proteins that increase with aging but are antagonized in physically active subjects (GESTALT, n = 58) and MA group. Enriched pathways from 40 proteins that decrease with PA and MA and increase with Age are shown (top), and pathways from 141 proteins that are in common between PA and Age (left) and enriched pathways from 80 proteins that are in common between MA and Age (right) are shown. Network analysis and enrichment analysis are performed using STRING analysis tool; the top enriched pathways are color coded (p<0.05). Proteins without interaction partners are omitted from visualization. (**b**) The subset of 162 MA- proteome represents cluster of chromatin organization, nuclear pore, mRNA splicing process, and contractile fiber proteins. This cluster of proteins appear unrelated to physical activity and may reflect unique biology in the MA group. Protein-protein interaction pathways and GO cellular components shown here are significantly enriched (p<1.0e-16).

were globally underrepresented in MA. These data are consistent with previous suggestions that the production of alternative splicing variants of structural and regulatory proteins is an integral part of the preprogrammed resilience strategies aimed to counteract drifts toward loss of function and damage accumulation (*Ferrucci et al., 2022*), such as those that follow the decline of energy availability secondary to mitochondrial impairment (*Bhadra et al., 2020*).

## Mitochondria and aging skeletal muscle

Mitochondria have long been implicated in aging biology in general, including skeletal muscle aging. Mitochondrial derangement may contribute to functional decline with aging though various mechanisms, including but not limited to reduced energy availability for contraction and other essential cellular activities,

increased production of ROS, inflammatory signaling, and release of $Ca^{2+}$ and activation of caspase 3 consequent to an event known as mitochondrial permeability transition (*Hepple, 2016*). In addition, fragments of mitochondrial membrane or mtDNA may trigger the NLRP3 inflammasome and contribute to local and systemic inflammaging (*Pereira et al., 2019*). Hence, preserving skeletal muscle mitochondrial function is a central mechanism for maintaining skeletal muscle health with aging.

Our analysis found mitochondrial proteins that cover a large variety of locations and functions were more abundant in highly functioning octogenarians than controls, including signaling proteins that fine-tune mitochondrial dynamics, mitochondrial biogenesis, TCA cycle, and respiratory electron transport. Evidence in the literature suggests that most of these changes are a consequence of higher physical activity (*Ubaida-Mohien et al., 2019a*). In keeping with this view, we recently showed that a reduction in intrinsic mitochondrial respiratory capacity (respiration normalized to the abundance of a complex III subunit) was only seen in very sedentary but not in physically active septuagenarian men (*Spendiff et al., 2016*), and data from the literature show that low physical activity rather than aging per se causes an increase in skeletal muscle mitochondrial ROS emission (*Gram et al., 2015*). Although we can only speculate on this point, one contributing factor to the higher abundance of mitochondrial proteins in MA may relate to mitochondrial adaptations incurred at the initiation of training in the MA group. Noting that the athletes in our MA group started training between 55 years of age (endurance athletes) and 65 years (sprint/power athletes) (see *Table 2*), the nature of the mitochondrial adaptations was likely in excess of the mild age-related impairment that would have been present at the age training was initiated. Thus, perhaps this training built in a 'buffer,' such that even similar rates of age-related decrements in mitochondrial proteins between both MA and NA would still yield the higher levels of mitochondrial proteins that we observed in MA versus NA at the participants' age when the muscle was sampled. Unfortunately, the cross-sectional nature of our study limits conclusions regarding this and other possibilities.

The mechanisms by which physical activity attenuates oxidative stress are complex and not completely understood. On the one hand, the promotion of autophagy and mitochondrial biogenesis jointly contributes to the recycling of damaged mitochondria and subsequent replacement with healthy mitochondria that are less likely to generate excessive ROS. On the other hand, exercise in MA likely upregulates enzymatic antioxidants such as SOD2 by an NRF2-KEAP1 mechanism (*Gao et al., 2020*). Although our proteomic analysis did not detect PGC-1α, we did observe higher levels in MA skeletal muscle for *PGC-1 and ERR-induced regulator in muscle protein 1* (PERM1), a regulator of mitochondrial biogenesis (*Cho et al., 2016*). In addition, the mitochondrial SIRT3 was elevated in MA muscle. Exercise activates SIRT3 by an AMP-activated protein kinase-dependent mechanism (*Brandauer et al., 2015*), and, in turn, SIRT3 deacetylates the mitochondrial antioxidant enzyme SOD2 boosting its ROS-scavenging activity (*Tao et al., 2010*). This is consistent with a previous study where they also reported a significantly higher level of SIRT3 and SOD2 in the skeletal muscle of master athletes (~15 years younger than studied here) compared to age-matched controls (*Koltai et al., 2018*). Finally, mitochondrial permeability transition is an important source of elevated mitochondrial ROS in skeletal muscle (*Burke et al., 2021*) and SIRT3, which was elevated in MA, reduces mitochondrial permeability transition by deacetylation of cyclophilin D (*Hafner et al., 2010*).

The differential representation of proteins that fine-tune mitochondrial dynamics between MA and NA is particularly interesting because an optimal dynamic balance of expression between pro-fusion (OPA1, MFNs) (*Tezze et al., 2017*) and pro-fission (DRP1) processes (*Dulac et al., 2020*) is essential for mitochondrial health. Consistent with this idea, our data showed a higher abundance of pro-fusion (OPA1, MFN2) and pro-fission (OMA1) proteins, as well as a higher abundance of mitochondrial electron transport complex assembly proteins (NUBPL, COA1, ACAD9, etc.) in MA donors. Collectively, the higher abundance of these proteins in MA suggests better maintenance of processes involving mitochondrial dynamics and cristae remodeling in MA. In addition, maintained mitochondrial dynamics is also likely conducive to the better preservation of mtDNA that we observed in MA, given the importance of mitochondrial dynamics to mtDNA integrity (*Bess et al., 2012*). Importantly, our proteomics data are consistent with phenotypic data showing a higher abundance of respiratory competent muscle fibers and higher mtDNA copy number in MA versus NA. Interestingly, there were eight mitochondrial proteins that had a lower abundance in MA than would be expected by random chance. Of these, four proteins were associated with GO Biological Process lipid biosynthesis (CYB5R3, CDS2, ACSL3, and STARD7). CDS2 is an essential intermediate in the synthesis of phosphatidylglycerol, cardiolipin, and phosphatidylinositol (PI), an important regulator of lipid storage (*Qi et al., 2016*). STARD7 is an intramitochondrial lipid transfer protein for phosphatidylcholine. These data are generally consistent with a recent magnetic resonance spectroscopy analysis of aging human

muscle, showing that elderly subjects who did not exhibit significant muscle atrophy had lower levels of skeletal muscle phospholipids (*Hinkley et al., 2020*). The other four proteins that had lower expression in MA were PGAM5 (regulator of mitochondrial dynamics), dimethylarginine dimethylaminohydrolase 1 (DDAH1), SHC1 (mitochondrial adapter protein), and WDR26 (negative regulator in MAPK signaling pathway). The significance of the lower expression of these proteins in MA is unclear. Of note, most of these proteins are primarily located in the endoplasmic reticulum, although they are also found in mitochondria. WDR26 is expressed mostly during mitochondrial stress and hypoxia, which is less likely to occur in MA compared to controls. The underrepresentation in MA of STARD7, a protein important to efficient phosphatidylcholine import by mitochondria as well as mitochondrial function and morphogenesis, may be considered counterintuitive. However, recent studies have suggested that STARD7 is a candidate effector protein of ceramide, a lipid known for its ability to initiate a variety of mitochondria-mediated cytotoxic effects. Thus, the underrepresentation of STARD7 in MA may be beneficial in this latter context (*Bockelmann et al., 2018*).

In summary, our data suggest that overrepresentation of mitochondrial quality control proteins and mitochondrial dynamics proteins in octogenarian MA muscle likely translates to better maintenance and remodeling of mitochondrial cristae, with higher energy availability that positively affects cellular adaptation to stress, and better maintenance of muscle metabolism.

## RNA splicing, nuclear pore complex, and aging

An upregulation of alternative splicing is commonly seen with aging in a variety of tissues that include skeletal muscle in both animal models (*Rodríguez et al., 2016*) and aging humans (*Ubaida-Mohien et al., 2019b*). Interestingly, after accounting for physical activity and age, we previously found that higher mitochondrial oxidative capacity as measured by $^{31}$P-spectroscopy was associated with upregulation of spliceosome proteins (*Adelnia et al., 2020*), which we have interpreted to suggest that upregulation of alternative splicing may represent a resilience response to confer benefits to mitochondrial function and thus limit the negative effects of aging (*Ferrucci et al., 2022*). On this basis, we hypothesized that highly functioning octogenarians would require less of upregulation of this resilience mechanism. Consistent with this idea, 80 proteins involved with mRNA splicing, metabolism of RNA, capped intron containing pre-RNA, and transcription coregulator activity that were shown previously to increase with aging (*Rodríguez et al., 2016*; *Ubaida-Mohien et al., 2019b*) were globally underrepresented in MA compared to controls. On this basis, we suggest that the lower representation of spliceosome proteins that regulate alternative mRNA splicing in MA may indicate that such compensatory upregulation of alternative splicing is not required in these individuals because their mitochondrial biology is better protected by other means (e.g., physical activity). Future analysis of RNA expression to examine expression of splice variants in MA versus NA would be important to further evaluate this premise.

An unexpected finding was that NPC proteins were less represented in MA than controls. The NPC proteins are involved in mRNA splicing regulation (*Stewart, 2019*), and therefore underrepresentation of NPC proteins in MA may be part of a global downregulation of splicing. In this respect, since posttranslational oxidative modification and activation of mitochondrial-mediated apoptotic pathways are associated with upregulation of NPC proteins (*Lindenboim et al., 2020*), a higher level of mitochondrial function in MA probably requires less protein turnover and thus less synthesis of nucleoporins and structural proteins. This idea is consistent with the discordant responses of mitochondrial versus non-mitochondrial ribosomal proteins, where we observed that 38 mitochondrial ribosomal proteins from 28S and 39S were significantly higher expressed in MA, whereas cytoplasmic ribosomal proteins (RPS2, RPLP0) were less abundant in MA.

Proteostasis maintenance pathways are important for skeletal muscle as components of myofibers are often damaged and must be replaced regularly. Proteins involved in proteostasis appear to have no single trend of change in MA octogenarians, with some chaperones and autophagy proteins underrepresented and some overrepresented in MA. It is possible that the long-term physical activity adaptation of the skeletal muscle in MA reduces the need for replacement of muscle proteins, for example, the higher fidelity of their mitochondria coupled with the higher expression of antioxidant proteins such as SOD2 may limit post-translational damage to proteins, thereby reducing the rate at which they need to be replaced.

## Evidence for factors other than exercise in MA proteome

To a large extent, the results of this study are consistent with the well-established benefits of exercise on mitochondrial and skeletal muscle health (*Hood et al., 2019*). However, the individuals we studied were world-class athletes in their 80s and it is unlikely that their high function can be accomplished by the majority

of older people, even assuming that they adhere to a strict exercise regimen. We expect that a fortuitous combination of genetics and environmental factors beyond exercise per se make them *winners*. Consistent with the idea that factors beyond those linked to physical activity contribute to such an extreme phenotype, we observed several mitochondrial-related proteins that were uniquely upregulated in MA versus normal aging, and several proteins involving RNA processing and the inflammasome that were uniquely downregulated in MA versus normal aging. As these proteins are not among those previously identified as exercise-responsive (*Ubaida-Mohien et al., 2019a*), we refer to these as the MA-specific proteome (see *Figures 6 and 7*). Although our MA cohort is too small to permit assessment of genetic/hereditary factors in these protein differences, our results identify important candidate protein pathways to explore for antiaging effects and suggest additional studies with larger numbers of subjects (and including other types of athletes) would be worthwhile.

In conclusion, our data underscore that mitochondrial pathways are key to maintaining a high level of physical function in advanced age. Furthermore, our data show that high physical function is also associated with preventing the general increase with aging in NPC proteins and spliceosome proteins. Whereas many of the differentially represented proteins in MA overlap with those affected by daily physical activity, we also identified several proteins that typically change with aging and were uniquely countered by MA but not by physical activity. The study of these unique proteins may reveal mechanisms that allow sporadic individuals to maintain high level of physical activity late in life, and understanding these mechanisms may indicate new therapeutic strategies for attenuating sarcopenia and functional decline with aging.

## Acknowledgements

We thank the Master Athlete study participants and the GESTALT participants. We thank Lauren Brick for assistance with figure design. Funding for this study was provided by operating grants from the Canadian Institutes of Health Research (MOP 84408 to TT and MOP 125986 to RTH). This work was supported in part by the Intramural Research Program of the National Institute on Aging, NIH, Baltimore, MD, USA.

## Additional information

### Funding

| Funder | Grant reference number | Author |
| --- | --- | --- |
| Canadian Institutes of Health Research | MOP 125986 | Russell T Hepple |
| Canadian Institutes of Health Research | 84408 | Tanja Taivassalo |
| National Institute on Aging | | Luigi Ferrucci |

The funders had no role in study design, data collection and interpretation, or the decision to submit the work for publication.

### Author contributions

Ceereena Ubaida-Mohien, Data curation, Formal analysis, Investigation, Methodology, Writing – original draft, Writing – review and editing; Sally Spendiff, Data curation, Formal analysis, Investigation, Methodology, Writing – review and editing; Alexey Lyashkov, Norah J MacMillan, Marie-Eve Filion, Data curation, Investigation, Writing – review and editing; Ruin Moaddel, Data curation, Investigation, Methodology, Writing – review and editing; Jose A Morais, Investigation, Performed muscle biopsies, Writing – review and editing; Tanja Taivassalo, Conceptualization, Funding acquisition, Supervision, Writing – review and editing; Luigi Ferrucci, Conceptualization, Funding acquisition, Writing – original draft, Writing – review and editing; Russell T Hepple, Conceptualization, Funding acquisition, Investigation, Supervision, Writing – original draft, Writing – review and editing

### Author ORCIDs

Ceereena Ubaida-Mohien ⓘ http://orcid.org/0000-0002-4301-4758
Luigi Ferrucci ⓘ http://orcid.org/0000-0002-6273-1613

Russell T Hepple http://orcid.org/0000-0003-3640-486X

### Ethics

Human subjects: Human subjects research was done with prior approval from the Institutional Review Board of the Faculty of Medicine at McGill University (A08-M66-12B) and according to the Declaration of Helsinki. All subjects provided written informed consent.

### Decision letter and Author response

Decision letter https://doi.org/10.7554/eLife.74335.sa1
Author response https://doi.org/10.7554/eLife.74335.sa2

## Additional files

### Supplementary files

- Transparent reporting form
- Supplementary file 1. MA Protein List and Fold Change Statistics.
- Supplementary file 2. Underrepresented and Overrepresented Proteins in MA.
- Supplementary file 3. MA Exclusive Proteome.

### Data availability

The mass spectrometry proteomics data have been deposited to the MassIVE with the dataset identifier MSV000086195(https://massive.ucsd.edu/ProteoSAFe/dataset.jsp?accession=MSV000086195).

The following dataset was generated:

| Author(s) | Year | Dataset title | Dataset URL | Database and Identifier |
|---|---|---|---|---|
| Ubaida-Mohien C, Spendiff S, Lyashkov L, Moaddel R, MacMillan NJ, Filion M-E, Morais JA, Taivassalo T, Ferrucci L, Hepple RT | 2020 | Unbiased proteomics, histochemistry, and mitochondrial DNA copy number reveal better mitochondrial health in muscle of high functioning octogenarians | https://massive.ucsd.edu/ProteoSAFe/dataset.jsp?accession=MSV000086195 | MassIVE, MSV000086195 |

The following previously published dataset was used:

| Author(s) | Year | Dataset title | Dataset URL | Database and Identifier |
|---|---|---|---|---|
| Ubaida-Mohien C, Gonzalez-Freire M, Lyashkov A, Moaddel R, Chia CW, Simonsick E M, Ferrucci L | 2019 | Proteomics of Human Skeletal Muscle | http://proteomecentral.proteomexchange.org/cgi/GetDataset?ID=PXD011967 | ProteomeXchange, PXD011967 |

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

## Appendix 1

### Histochemical labeling for respiratory compromised muscle fibers

COX/SDH histochemistry (*Old and Johnson, 1989*; *Taylor et al., 2003*) was performed to assess the activity of OXPHOS complexes IV (COX) and II (SDH), and thus identify muscle fibers with a respiratory chain deficiency (COX[Neg]). The COX incubation medium was prepared by adding 100 µM cytochrome *c* to 4 mM of 3,3-diaminobenzidine tetrahydrochloride (DAB) with 20 µg of catalase. The slides were incubated for 45 min at 37°C in a humidified chamber. Following three washes in PBS, SDH incubation medium (130 mM sodium succinate, 200 µM phenazine methosulfate, 1 mM sodium azide, and 1.5 mM NitroBlue tetrazolium) was added to the sections. The sections were again incubated for 45 min at 37°C, washed 3× PBS, and then dehydrated through a graded ethanol series and xylene before being mounted in DPX. Images of the whole-muscle section were captured on a Zeiss Axio Imager M2 fluorescent microscope (Zeiss, Germany). Counts of COX-positive (COX[Pos]), COX[Int], and COX[Neg] myofibers were performed for the whole-muscle cross-section. COX-negative fibers are indicative of cells with high levels of mtDNA mutations (*Bua et al., 2006*) and will thus not demonstrate the brown reaction product (oxidized DAB) during the first incubation but will stain blue following the second incubation for SDH activity. This is because the nuclear DNA entirely encodes SDH, so any mtDNA mutations will not affect its activity. In contrast, mtDNA mutations could affect complex IV activity and prevent DAB oxidation if a mutation affects a region of mtDNA containing the Cox subunit genes.

### Mitochondrial DNA copy number

Groups of 25 fibers (5 × 5 fibers) in an unstained 20-µm-thick muscle cross-section were randomly selected (random number generator and numbered grid), laser captured, and their DNA extracted using the lysis method (*Spendiff et al., 2013*) and stored at –20°C. The products were then separated on a 1% agarose gel containing Sybr Safe DNA Gel Stain at 70 V for 30 min, and the bands visualized using a G-Box chem imaging system (*Figure 1—figure supplement 1a*). The mtDNA fragment was extracted using a QIAquick Gel Extraction Kit and quantified using a NanoDrop-2000 spectrophotometer (Thermo Scientific). Total mtDNA copy number in muscle fibers was determined using a standard curve created from the amplification of MTND1 (1011 bp fragment, forward primer: 5′ TGTAAAACGACGGCCAGT 3′, reverse primer: 5′ CAGGAAACAGCTATGACC) (*Greaves et al., 2010*; *Figure 1—figure supplement 1b*). The products were separated on a 1% agarose gel, and the mtDNA fragment extracted using a QIAquick Gel Extraction Kit and quantified with a NanoDrop-2000 spectrophotometer (Thermo Scientific). A standard curve was generated by serially diluting down the sample in $dH_2O$. Groups of 25 fibers (5 × 5 fibers) were randomly selected (random number generator and numbered grid) and laser captured. Samples along with the standard curve were run in triplicate using an MTND1 TaqMan qPCR assay (forward primer: 5′ CCCTAAAACCCG CCACATCT 3′, reverse primer: 5′ GAGCGATGGTGAGAGCTAAGGT 3′, probe: 5′ VIC-CCATCACC CTCTACATCACCGCCC 3′). The total mtDNA copy number was then determined using the sample $C_q$ values and the equation generated from the standard curve. Results were divided by the total area of the captured cells to give mtDNA copy number per area.

### Western blotting for mitochondrial proteins

Western blotting for representative mitochondrial proteins was performed as described previously (*Spendiff et al., 2016*). Briefly, 10–20 mg of muscle was homogenized in a Retch mixer mill (MM400) with 10× (w/v) of extraction buffer (50 mM Tris base, 150 mM NaCl, 1% Triton X-100, 0.5% sodium deoxycholate, 0.1% sodium dodecyl sulfate), and 10 µl/ml of Protease Inhibitor Cocktail. Following 2 hr of gentle agitation at 4°C, samples were centrifuged at 12,000 × *g* for 20 min at 4°C, and the supernatant removed for protein assessment by Bradford assay. Samples were diluted in 4× Laemmli buffer to yield a final protein concentration of 2 µg/ml and then boiled for 5 min at 95°C. Immunoblotting was done using 20 µg of protein, loaded onto a 12% acrylamide gel, electrophoresed by SDS-PAGE and then transferred to polyvinylidene fluoride membranes (Life Sciences), blocked for 1 hr at room temperature in 5% (w/v) semi-skinned milk, and probed overnight at 4°C with the following primary antibodies (diluted in 5% BSA): mouse monoclonal anti-VDAC (1:1000; Abcam ab14734) and mouse monoclonal Total OXPHOS Cocktail (1:2000, Abcam ab110413). To address the poorer sensitivity to the CIV subunit in this cocktail after boiling human samples, we also probed using mouse monoclonal CIV (1:1000, Life Technologies A21348). Ponceau staining was performed to normalize protein loading. Following washing, membranes were incubated with HRP-conjugated secondary antibodies (diluted in 5% milk, Abcam) for 1 hr at room temperature. Protein bands

were detected using SuperSignal West Pico Chemiluminescent Substrate (Thermo Scientific, USA) and imaged with a G-Box Chem Imaging System. Analysis of protein bands was performed using GeneTools software (Syngenem, UK).

## Sample preparation and protein extraction for MS

Roughly, 5–8 mg of vastus lateralis muscle tissue was pulverized in liquid nitrogen and mixed with the modified SDT lysis buffer (100 mM Tris, 140 mM NaCl, 4% SDS, 1% Triton X-114, pH 7.6; Sigma) (*Wiśniewski et al., 2009*). Tissues were sonicated using preprogrammed tabletop tip sonicator, centrifuged at +4°C for 15 min at 14,000 rpm, aliquoted, and stored at –80°C until further processing. Protein concentration was determined using a commercially available 2-D quant kit (GE Healthcare Life Sciences). The sample quality was confirmed using NuPAGE protein gels stained with fluorescent SyproRuby protein stain (Thermo Fisher). 300 µg of muscle tissue lysate was used for tryptic digestion.

Detergents and lipids were removed by standard methanol/chloroform extraction protocol (sample:methanol:chloroform:water – 1:4:1:3) (*Wessel and Flügge, 1984*). Purified proteins were resuspended using a small aliquot (30 µl) of concentrated urea buffer (8 M urea, 2 M thiourea, 150 mM NaCl; Sigma), reduced with 50 mM DTT for 1 hr at 36°C, and alkylated with 100 mM iodoacetamide for 1 hr at 36°C in the dark. Concentrated urea was diluted 12 times with 50 mM ammonium bicarbonate buffer. Proteins were digested for 18 hr at 36°C using trypsin/LysC mixture (Promega) in 1:50 (w/w) enzyme to protein ratio. Protein digests were desalted on 10 × 4.0 mm C18 cartridge using Agilent 1260 Bio-Inert HPLC system connected to the fraction collector. Purified peptides were speed vacuum dried and stored at –80°C.

Initially, three independent 10-plex TMT (TMT 10-plex) experiments were designed. Samples in each TMT experiment were blinded and randomized between TMT channels to avoid labeling and sampling bias. Each LC-MS experiment used 100 µg of muscle tissue digest from five MA samples matched with five controls (NA) that were independently labeled with 10-plex tags (Thermo Fisher). Of the three 10-plex experiment, a total of 24 biological replicates and 6 technical replicates were included, technical replicates were used to optimize instrument performance and to estimate technical reproducibility. 200 femtomole of bacterial beta-galactosidase digest (SCIEX) was spiked into each sample prior to TMT labeling to control labeling efficiency and overall instrument performance. Labeled peptides were combined into one experiment and fractionated.

## High-pH RPLC fractionation and concatenation strategy

Basic reverse-phase fractionation was done on Agilent 1260 Bio-Inert HPLC system as previously described (*Wang et al., 2011*). XBridge 4.6 mm × 250 mm column (Peptide BEH C18) equipped with 3.9 mm × 5 mm XBridge BEH Shield RP18 XP VanGuard cartridge (Waters). The solvent composition was as follows: 10 mM ammonium formate (pH 10) as mobile phase (A) and 10 mM ammonium formate and 90% ACN (pH 10) as mobile-phase B. Labeled peptides were separated using a linear organic gradient (5–50% B in 100 min). Initially, 99 fractions were collected during each LC run at 1 min intervals each. Three fractions separated by 33 min intervals were concatenated into 33 master fractions, as previously described (*Ubaida-Mohien et al., 2019a*).

## Nano LC-MS/MS analyses

Purified peptide fractions from muscle tissues were analyzed using UltiMate 3000 Nano LC Systems coupled to the Q Exactive HF mass spectrometer (Thermo Scientific, San Jose, CA). Each fraction was separated on a 45 cm capillary column with 150 µm ID on a linear organic gradient using 550 nl/min flow rate. Gradient went from 5% to 35% B in 195 min. Mobile phases A and B consisted of 0.1% formic acid in water and 0.1% formic acid in acetonitrile, respectively. Tandem mass spectra were obtained using Q Exactive HF mass spectrometer with the heated capillary temperature +280°C and spray voltage set to 2.5 kV. Full MS1 spectra were acquired from 330 to 1600 $m/z$ at 120,000 resolution and 50 ms maximum accumulation time with automatic gain control (AGC) set to 3 × 106. Dd-MS2 spectra were acquired using dynamic $m/z$ range with a fixed first mass of 100 $m/z$. MS/MS spectra were resolved to 30,000 with 150 ms of maximum accumulation time and AGC target set to 1 × 105. Fifteen most abundant ions were selected for fragmentation using 29% normalized high collision energy. A dynamic exclusion time of 70 s was used to discriminate against the previously analyzed ions.

## Proteomics informatics

The PeptideProphet model fits the peptide-spectrum matches into two distributions, one an extreme value distribution for the incorrect matches, and the other a normal distribution for correct matches. The protein was filtered at thresholds of 0.01% peptide FDR, 1% protein FDR, and requiring a minimum of one unique peptide for protein identification.

Single-peptide hits were allowed when any quantifiable peptide was detected across at least 30% of all samples (n = 24) and if proteins were identified with more than one search engine. Reporter ion quantitative values were extracted from Scaffold and decoy spectra, and contaminant spectra and peptide spectra shared between more than one protein were removed. Typically, spectra are shared between proteins if the two proteins share most of their sequence, usually for protein isoforms. Reporter ions were retained for further analyses if they were exclusive to only one protein, and they were identified in all 10 channels across each TMT batch. Further protein bioinformatics was performed, as previously described (*Ubaida-Mohien et al., 2019a*).

