## [Editor Report]

Proteomics studies of skeletal muscle biopsies in healthy individuals demonstrate that older age was associated with an underrepresentation of mitochondrial proteins, especially those associated with oxidative phosphorylation and energy metabolism. Ubaida-Mohien et al. analyzed muscle protein differences between octogenarian master athletes and non-athletes demonstrating that high muscle function during aging is associated with the preservation of structural and functional proteins in mitochondria such as electron transport capacity, cristae formation, mitochondrial biogenesis, and mtDNA-encoded proteins. The authors propose that the study of these unique proteins may uncover molecular mechanisms to design therapeutic strategies for skeletal muscle functional decline with aging.

---

## [Decision Letter]

**Decision letter after peer review:**

Thank you for submitting your article "Unbiased Proteomics, Oxphos Histochemistry, and mtDNA Copy Number Reveal Better Mitochondrial Health in Muscle of High Functioning Octogenarians" for consideration by *eLife*. Your article has been reviewed by 2 peer reviewers, and the evaluation has been overseen by a Reviewing Editor and Carlos Isales as the Senior Editor. The following individuals involved in review of your submission have agreed to reveal their identity: Mathew Piasecki (Reviewer #1); Hans Degens (Reviewer #2).

Essential revisions:

The reviewers noted the importance of this work with an extremely rare human cohort that offers new insights into the aged human muscle proteome. However, some issues were identified that may require further experimental support and/or explicit discussion in the text.

1. Modify the term "preservation" to "training-induced adaptations" of mitochondrial structure/function proteins. It is well known that even those who begin training later in life can still make gains in maximal oxygen uptake and muscle mass and strength. In addition, athletes also seem to have started competition quite late in life.

2. Provide measurement of VO2max per kg of lean mass and peak cycle workrate.

3. Provide a justification for why there is greater muscle size in MA, although there is no difference in fiber size and fiber type proportion. It may be appropriate to plot muscle CSA against the average size of all fibers combined.

4. Provide VL CSA rather than Thigh CSA normalized to height.

*Reviewer #1 (Recommendations for the authors):*

In addition to points above, please see several more specific comments below, which I hope the authors will find useful.

Line 70 – This is over simplistic and does not prove that exercise prevents commonly observed effects of ageing. I appreciate the argument the authors are trying to make, but I have no doubt those discussed in the cited article have also deteriorated with age, as all masters athletes have. I would suggest a minor rewording to highlight why you view the older athletes to be an entirely different population.

Line 77 – This is not a widely accepted finding (MU number and NMJ trans). It would be beneficial to highlight opposing findings where relevant, particularly in relation to extremely high NF Jiggle values (e.g. compared to https://pubmed.ncbi.nlm.nih.gov/25727901/, https://physoc.onlinelibrary.wiley.com/doi/full/10.14814/phy2.12987).

Line 134 – stand up and go, or timed up and go (TUG)?

MA time competing seems low, they are not lifelong athletes. How would this influence the results?

Line 149 – here and elsewhere it may seem to some that the authors are hoping for differences that are not apparent. No doubt something we are all guilty of bud terms such as this should be avoided.

Figure 2.2 Please report VL CSA rather than Thigh CSA normalised to height. Is this full quads or VL only? These data are useful and there is no reason why normalising to height is superior to the commonly reported CSA in recognised units.

The MRI image is poor quality, (g) is elongated.

Line 81 – Using a discovery?

Line 220 – What is a respiratory compromised fibre? Distinct definition?

Line 248 – Does the lower abundance of RPS2 and RPLP0 support the preceding statement that mito synthesis is higher in MA.

Line 324 – What is the difference between less abundant and significantly less abundant?

Line 328 – A lower p value does not necessarily mean a greater magnitude of difference.

Line 417 – massive may be viewed as subjective.

Line 558 – The athletes also appear to have taken up competition fairly late in life and would be considered 'old' prior to undertaking any competition. This is an important point and it would be useful to address in the discussion. More specifically, is it logical to assume the MA have reversed mitochondrial impairments associated with age?

Line 602 – muscle mass by MRI? If this refers to muscle volume then please include.

Line 606 – please clarify here what constitutes a respiratory compromised fiber (link to section 4.5 perhaps)

Line 683 – "with p<0.05"?

Line 685 – please clarify factors in ANOA.

*Reviewer #2 (Recommendations for the authors):*

Overall a good study indeed with interesting data that are well presented. I have some issues that I hope the authors will consider.

In the Introduction first paragraph I have some reservation about the comment 'However, there is clear evidence that the "usual" decline of strength and function is not escapable…'. In fact, the decline in master athletes is in absolute terms even larger than in non-athletes. Consider e.g. the age-related decline in VO2max (Tanaka and Seals, J Physiol, 2008) and muscle power (Pearson et al., MSSE, 2002) that are actually the same % wise and in absolute terms even larger than in non-athletes (note in these papers that the age-related decline in VO2max and power in athletes and controls converge). However, training at any age of non-athletes may get them up to the level of the master athletes, as again Hepple et al., 1997 have shown that both VO2max and muscle strength can be increased even in older people. Also, consider that in the paper by Power GA et al., (MSSE 2010) the older master athletes were in fact weaker than the older non-athletes! Thus, the comment that the decline is escapable is in my view not correct. Just rephrase to something along the line of 'At any age, master athletes have better performance than non-athletes'.

Results: Interesting that you had 8 women and 7 men, where the participation in master athletics is higher in men than women. Nothing for you to comment on, just find it curious.

In figure 1, could you also add a figure of VO2max per kg lean mass, as it is the lean mass that realises the VO2max. You can do this, as you have the fat% in table 1.

Use the same symbols in Figure 1i as in the other panels.

Page 7 top: If there is no difference in fibre size, and no difference in fibre type proportion, what is then the cause of the larger size of the muscles of the MA? Is there perhaps attenuated loss of fibres? Can you perhaps plot muscle CSA vs average size of all fibres pooled? In Figure S1 C both type I and type 2 fibres are not significantly higher in MA than NA, but perhaps when you do the same analyses for the FCSA of all fibres pooled you find a significantly larger fibre size in the MA.

The overlapping circles in figure 2a are somewhat misleading as the area of overlap of all three circles is the largest are, yet depicted in the circles as the smallest. Perhaps you can present it differently to make this clearer, but I understand the complication as the non-overlap region may become so tiny to become for practical purposes invisible. See, if you can, how it pans out.

Page 9 line 194: Perhaps we can make a start to get rid of the common misnomer 'energy production' as what is really meant 'ATP production'. The first law of thermodynamics states that energy can not be generated, though I accept it is common parlour (incidentally, that is also why I have a problem with the term 'renewable energy').

I notice in figure 1 and figure 3 that there is overlap between the NA and MA with some NA being better off than the lowest MA. Perhaps you can capitalise on this by analysing relationship between parameters in figure 1 with parameters in figure 3 (those in panel d,f,h,i).

Page 11 line 221: I find the term 'compromised' here and elsewhere in the paper (also in the abstract I now reading this realise) misleading. This is as 'compromise' suggests 'defective' but you showed no evidence of defective function, but rather some fibres have a higher COX activity than other fibres. This is a common observation in muscles from young people also, and we would not call the checkerboard pattern of muscle fibres in the young indicative for fibres with 'compromised' mitochondrial function. It thus also are not 'competent'(line 223) or 'incompetent' fibres, but just fibres with a lower or higher COX activity. However, if you mean to say COX-/SDH^+^ then you are justified, but then you need to clarify that in figure 3f x-axis and identify them as such, and not just COX-.

Page 11 line 233-245: You conclude that the higher mtDNA number in MA than NA is supportive of a lower presence of mtDNA mutations. This is a non-sequitur! You have more mtDNA because you have more mitochondria; this says nothing about mutation load. Please remove this speculation in the opening sentence and as conclusion from this paragraph.

Page 12 line 252-266: The altered MICOS may help respiratory capacity in the MA. I accept that. But this is a useful adaptation, and the lower values in NA does not indicate compromised mitochondrial function, but rather that the mt do not need such high cristae density to generate the lower amount of ATP that is needed by the NA than the MA. So, also here, please refrain from using the term 'compromised'. I would accept such a term if you had measured e.g. larger leakage or decoupling. Incidentally, the decrease in abundance of the mentioned proteins in ageing may likewise be a reflection of the age-related reduction in physical activity, where disuse is well-known to reduce the muscle oxidative capacity.

Consider to increase the font in figure 4b; I could only read it when having the figure very close by and straining my eyes.

Page 14 line 333-341: I don't see the logical link between changes in the spliceosome pathway and mitochondrial function. Please explain why they are, according to you, linked. Circumstantial observations that Mt are up and spliceosome pathways are down in the MA is just that: circumstantial. You can then not say, as that is circular reasoning, that therefore 'a resilience response (that is higher spliceosome) is not required in athletes because of high mt'; that is explaining the observation by the observation.

I like the paragraph on page 16, where you disentangle physical activity and ageing.

Page 17: Nice comparison. Do, however, consider to change the term 'affected' by 'higher' or 'lower' where appropriate, as it is that you have measured.

Consider the above points also for the discussion. Having said that, the first three paragraphs give a good summary and interpretation of the findings with which I can agree.

On page 20 on mitochondria: Please make sure you realise you have not collected evidence of mitochondrial derangement in old age, or prevention thereof in MA, but rather that the abundance of mt proteins was higher. So, in the writing of this part of the discussion do not be tempted to put more into it than you can say based on your data. Note that fewer mitochondria does not mean loss of mitochondrial function; glycolytic fibres in young people -with few mitochondria- function perfectly well!

Page 24 560-563: Probably it is the case that part of the success is related to the genetics of the athletes. However, this may well be a minimal factor, as neither years of training, nor weekly duration of training are related to performance. In addition, there is evidence that people that performance at the age of 80 did not differ significantly between those who had continued competing till at least 85 vs those who had stopped after 80. This observation suggests that there is no selection of the best performers remaining competitive. In addition, again Hepple at al., 1997, but also people as Fiatarone et al., 1990 (training in nonagerians) and Harridge et al., M and N, 1999 show that people even at the age of people in this study can still be trained. I therefore believe that genetics plays only a minor role.

The conclusion is justified, though, as I think you will gather by now, I have reservations about this statement in the conclusion that 'that allow sporadic individuals to maintain high levels of physical activity', as the point is that anybody who wants to put in the effort, even in old age, can induce the proteins seen in the MA and as has been shown repeatedly (see refs above) improve their function. So, the main changes observed are a reward for their training, and the therapy suggested is therefore 'train', or at least remain physically active.

---

## [Author Response]

Essential revisions:The reviewers noted the importance of this work with an extremely rare human cohort that offers new insights into the aged human muscle proteome. However, some issues were identified that may require further experimental support and/or explicit discussion in the text.1. Modify the term "preservation" to "training-induced adaptations" of mitochondrial structure/function proteins. It is well known that even those who begin training later in life can still make gains in maximal oxygen uptake and muscle mass and strength. In addition, athletes also seem to have started competition quite late in life.

We thank the Editor and Reviewer for the comment on this issue, however, we wish to point out that whilst there is ample data showing muscle plasticity in 65-75 y old humans, to our knowledge there are no data pertaining to muscle adaptations to endurance training in octogenarians that we are aware of. Animal model data, on the other hand, indicates that skeletal muscle has a markedly impaired ability to adapt to endurance training in advanced age (the studies cited below modeled a similar relative age to a 65 y old and an octogenarian human based upon speciesspecific survival curve data). Specifically, whereas initiating treadmill exercise training in 28 mo old rats (models approximately 65 y old human based on survival curve data for the Fisher 344 x Brown Norway F1 hybrid rat) yields an improved skeletal muscle VO2peak and mitochondrial enzyme activities after 2 months of training (Betik et al., Exp Physiol. 93[7]: 863-71, 2008), continuing that training for a further 5 months (rats are now 35 mo old which models approximately 80 y of age in humans) shows no benefit to skeletal muscle VO2peak or mitochondrial enzyme activities in the trained relative to age-matched sedentary rats (Betik et al., Am J Physiol Regul Integr Compar Physiol. 297[3]: R744-55, 2009) and age-related skeletal muscle atrophy and oxidative protein damage were both exacerbated by the endurance training (Thomas et al., Exp Gerontol. 45[11]: 856-67, 2010), despite other systemic benefits of endurance exercise such as improved lifespan, and reduced cardiomyocyte loss and cardiac fibrosis being evident in the trained rats (Wright et al., Exp Gerontol. 50: 9-18, 2014). The data pertaining to resistance training in octogenarian humans, notwithstanding work from Harridge and others showing improvements in strength, also shows that the skeletal muscle in advanced age has a markedly impaired ability to adapt to resistance training when assessed at the single muscle fiber level. Specifically, Trappe’s group found that although there was an improvement in strength with resistance training, there was no change in fiber cross-sectional area, shortening velocity, peak tetanic force, or power when assessed in segments of type I or type II fibers obtained by biopsy from octogenarian men (Slivka et al., Am J Physiol Regul Integr Compar Physiol. 295[1]: R273-80-, 2008) and women (Raue et al., J Appl Physiol. 106[5]: 1611-17, 2009). On this basis, the fact that there were still improvements in strength with the training in Trappe’s prior studies of octogenarians (consistent with the work of Harridge and others) has been suggested to implicate motor control adaptations rather than intrinsic muscle fiber adaptations. There are also animal model studies showing diminished muscle adaptive signaling in response to contractile activity in skeletal muscle from rats at a similar relative age to octogenarian humans based on species-specific survival curves. For example, work from David Hood’s group reported reduced mitochondrial biogenesis signaling (p38, CaMKII, and AMPKalpha) in response to electrically evoked muscle contractions in skeletal muscle from very old rats (Ljubicic et al., Aging Cell 8[4]: 394-404, 2009). As such, our interpretation of the literature is that whilst skeletal muscle retains adaptive plasticity in 65-75 y old humans (as noted in reviewer two’s comments), skeletal muscle plasticity becomes impaired in advanced age (octogenarians). We specifically chose the word “preservation” to avoid making an a priori conclusion about the causes of the differences in mitochondrial proteins in MA vs NA. We agree that exercise is part of the equation, but we did not want to overlook other contributing factors. Indeed, our analysis shows (Figure 5) that 301 proteins that were differentially represented in MA did not overlap with proteins previously shown to be responsive to exercise training/physical activity, underscoring the point that not all of the differences in the muscle proteome seen in MA can be attributed to their physical activity. Even if we agree that training would have induced the usual adaptations in the MA at an earlier point in their aging trajectory (i.e., when they started training for competition), by the point in time when we were studying the athletes all had been competing and training for many years (see Table 2), meaning that any training-induced adaptation had occurred years prior. In this context, the higher levels of mitochondrial proteins seen in MA could be a combination of both attenuated age-related decline as well as training-induced maintenance, and in both cases we feel the term “preservation” is the most appropriate descriptor. We hope this clarifies the rationale for our use of the term “preservation” as opposed to “training-induced adaptations”. Notwithstanding this point, we have added text to lines 518-528 of the discussion to address the possible contribution of training in overcoming any mitochondrial impairment at the age when competitive training was iniated in the MA participants (see response to Reviewer 1, below).

2. Provide measurement of VO2max per kg of lean mass and peak cycle workrate.

We now provide the group values for VO2max per kg of lean mass in Figure 1 panel A. Data for peak cycle work rate was previously provided in Figure 1 panel B.

3. Provide a justification for why there is greater muscle size in MA, although there is no difference in fiber size and fiber type proportion. It may be appropriate to plot muscle CSA against the average size of all fibers combined.

As seen in the prior supplemental Figure S1C, there was a tendency to higher fiber cross-sectional area across type I, type IIa and hybrid fiber types in MA versus NA. We have now included mean fiber cross-sectional area for all fibers in this plot and note that the mean fiber cross-sectional area was 28% greater in MA (P=0.0245), which is very similar to the 30% higher vastus lateralis muscle cross-sectional area now shown in revised Figure 1i. These findings suggest the primary difference in muscle area between groups is accounted for by the difference in fiber size. We hope this clarifies our observations.

4. Provide VL CSA rather than Thigh CSA normalized to height.

We now provide vastus lateralis CSA in Figure 1i as requested.

Reviewer #1 (Recommendations for the authors):In addition to points above, please see several more specific comments below, which I hope the authors will find useful.Line 70 – This is over simplistic and does not prove that exercise prevents commonly observed effects of ageing. I appreciate the argument the authors are trying to make, but I have no doubt those discussed in the cited article have also deteriorated with age, as all masters athletes have. I would suggest a minor rewording to highlight why you view the older athletes to be an entirely different population.

We agree completely and realize our wording was misleading. We have revised as follows:

“However, there is clear evidence that the degree of such “usual” decline of strength and function is less severe in some individuals. For example, master athletes exhibit considerably higher physical performance capacity in their eighties and nineties than their sedentary counterparts …”

Line 77 – This is not a widely accepted finding (MU number and NMJ trans). It would be beneficial to highlight opposing findings where relevant, particularly in relation to extremely high NF Jiggle values (e.g. compared to https://pubmed.ncbi.nlm.nih.gov/25727901/, https://physoc.onlinelibrary.wiley.com/doi/full/10.14814/phy2.12987).

We appreciate your perspective on this point and note that the athletes studied in the Power 2016 article were the very same individuals as studied in the current investigation. In addition, these athletes were truly world class, with 8 of them being world record holders in their discipline and age category at the time of study, and the remaining individuals being ranked in the top 5 in the world for their age and discipline. This makes them of a higher quality (performance-wise) than the subjects studied by Dr. Piasecki and colleagues previously (Physiol Reports. 4[19]: e12987, 2016) and may explain the different results. Notwithstanding that possibility, our intention in mentioning the Power et al., study was to provide an example of how we have used study of highly functioning elderly to identify features that may contribute to their high function. It was not intended to represent the entirety of the field with respect to findings concerning motor unit numbers in masters athletes.

Line 134 – stand up and go, or timed up and go (TUG)?

Thank you for pointing out this error. We have now corrected the text and figure.

MA time competing seems low, they are not lifelong athletes. How would this influence the results?

Without doing a comparison, it is difficult to draw firm conclusions, but we note that our proteomics analysis suggests not all of the differences in proteins seen in MA could be attributed to established exercise-responsive proteins. As such, it seems likely that some of these non-exercise related differences would persist irrespective of when the training was initiated.

Line 149 – here and elsewhere it may seem to some that the authors are hoping for differences that are not apparent. No doubt something we are all guilty of bud terms such as this should be avoided.

We appreciate this point and have revised the wording.

Figure 2.2 Please report VL CSA rather than Thigh CSA normalised to height. Is this full quads or VL only? These data are useful and there is no reason why normalising to height is superior to the commonly reported CSA in recognised units.

We now report vastus lateralis cross-sectional area in Figure 1i.

The MRI image is poor quality, (g) is elongated.

This has been corrected.

Line 81 – Using a discovery?

Apologies for the error. The sentence has been corrected to “Using an unbiased discovery proteomics approach”.

Line 220 – What is a respiratory compromised fibre? Distinct definition?

We have now clarified the criteria used for judging respiratory competence versus compromise in this section (lines 222-231) as well as the Methods (see below).

Line 248 – Does the lower abundance of RPS2 and RPLP0 support the preceding statement that mito synthesis is higher in MA.

We apologize for the confusion, as the two statements are not related. Our data indicate that mitoribosome translation may be higher, while cytoplasmic ribosomes may be lower. We have now clarified our wording in the text.

Line 324 – What is the difference between less abundant and significantly less abundant?

We have corrected this statement to refer only to those proteins that were significantly different.

Line 328 – A lower p value does not necessarily mean a greater magnitude of difference.

We agree and have revised the wording to make the point clear.

Line 417 – massive may be viewed as subjective.

We have revised the statement to remove the qualitative element.

Line 558 – The athletes also appear to have taken up competition fairly late in life and would be considered 'old' prior to undertaking any competition. This is an important point and it would be useful to address in the discussion. More specifically, is it logical to assume the MA have reversed mitochondrial impairments associated with age?

The average age of the athletes was 80 y and based upon a range of nearly 27 y training history in the endurance athletes and 17 y in the sprint/power athletes (Table 2), this would make the athletes approximately 55 y at the start of the competitive phase for endurance athletes and approximately 65 y for the sprint/power athletes. It is not clear to what degree mitochondrial impairments may have pre-existed in these individuals at the time they started training but it seems likely that any impairment would have been fairly mild because of their age at the time competitive training began.

Line 602 – muscle mass by MRI? If this refers to muscle volume then please include.

This was a typographical error and we have now corrected it.

Line 606 – please clarify here what constitutes a respiratory compromised fiber (link to section 4.5 perhaps)

This has now been clarified as requested (P. 26, lines 631-645).

Line 683 – "with p<0.05"?

We have revised for clarity.

Line 685 – please clarify factors in ANOA.

Done as requested.

Reviewer #2 (Recommendations for the authors):Overall a good study indeed with interesting data that are well presented. I have some issues that I hope the authors will consider.In the Introduction first paragraph I have some reservation about the comment 'However, there is clear evidence that the "usual" decline of strength and function is not escapable…'. In fact, the decline in master athletes is in absolute terms even larger than in non-athletes. Consider e.g. the age-related decline in VO2max (Tanaka and Seals, J Physiol, 2008) and muscle power (Pearson et al., MSSE, 2002) that are actually the same % wise and in absolute terms even larger than in non-athletes (note in these papers that the age-related decline in VO2max and power in athletes and controls converge). However, training at any age of non-athletes may get them up to the level of the master athletes, as again Hepple et al., 1997 have shown that both VO2max and muscle strength can be increased even in older people. Also, consider that in the paper by Power GA et al., (MSSE 2010) the older master athletes were in fact weaker than the older non-athletes! Thus, the comment that the decline is escapable is in my view not correct. Just rephrase to something along the line of 'At any age, master athletes have better performance than non-athletes'.

The reviewer raises an excellent point and we have revised the text to make it clear that we are not positing that the declines with aging can be completely prevented in some individuals, but rather that the various indices of physical function can be superior in master athletes. Although we acknowledge that there is no evidence either way, it seems very unlikely that any amount of training would be sufficient to shift the average sedentary octogenarian to a worldclass octogenarian track and field athlete (8 out of our 15 athletes were world record holders in their discipline for their age at the time of study, and the remaining were top 5 in their age group based on world rankings). Indeed, it is striking watching world masters competitions and seeing how much faster/better the top athletes are relative to the remainder of the competitors, yet presumably many of the slower athletes are also training as hard as they can. In this respect, we posit that the octogenarian athlete is similar to young athletes who are also the product of both their training and their intrinsic “talent” (top athletes are neither born nor made – you need both factors to win). Our point here is that we expect that training alone does not explain the performance of these individuals and there have to be other factors at play. We suspect that having a slower trajectory of aging biology that permits a better maintenance of body structure and function may be one of these factors (even if this is difficult to objectively disentangle from exercise). Notwithstanding, we propose that studying individuals who have extraordinary function in advanced age informs us about the mechanisms which might favor a better aging trajectory.

Results: Interesting that you had 8 women and 7 men, where the participation in master athletics is higher in men than women. Nothing for you to comment on, just find it curious.In figure 1, could you also add a figure of VO2max per kg lean mass, as it is the lean mass that realises the VO2max. You can do this, as you have the fat% in table 1.

Done as requested.

Use the same symbols in Figure 1i as in the other panels.

Done as requested.

Page 7 top: If there is no difference in fibre size, and no difference in fibre type proportion, what is then the cause of the larger size of the muscles of the MA? Is there perhaps attenuated loss of fibres? Can you perhaps plot muscle CSA vs average size of all fibres pooled? In Figure S1 C both type I and type 2 fibres are not significantly higher in MA than NA, but perhaps when you do the same analyses for the FCSA of all fibres pooled you find a significantly larger fibre size in the MA.

We now include this analysis in the revised supplemental Figure S1 and as suggested, we do find a significantly higher mean fiber cross-sectional area in MA versus NA. We have revised the text accordingly. On this basis, our data suggests larger mean fiber size is the main determinant of higher muscle mass in MA. Thank you for making this suggestion as it has certainly clarified the interpretation.

The overlapping circles in figure 2a are somewhat misleading as the area of overlap of all three circles is the largest are, yet depicted in the circles as the smallest. Perhaps you can present it differently to make this clearer, but I understand the complication as the non-overlap region may become so tiny to become for practical purposes invisible. See, if you can, how it pans out.

We have revised panel 2a and hope you find this a more logical representation of the data.

Page 9 line 194: Perhaps we can make a start to get rid of the common misnomer 'energy production' as what is really meant 'ATP production'. The first law of thermodynamics states that energy can not be generated, though I accept it is common parlour (incidentally, that is also why I have a problem with the term 'renewable energy').

We appreciate the reviewer’s point and have changed the text to read “ATP production”.

I notice in figure 1 and figure 3 that there is overlap between the NA and MA with some NA being better off than the lowest MA. Perhaps you can capitalise on this by analysing relationship between parameters in figure 1 with parameters in figure 3 (those in panel d,f,h,i).

Although this may be an interesting analysis, it is tangential to our purpose of comparing these two groups of individuals.

Page 11 line 221: I find the term 'compromised' here and elsewhere in the paper (also in the abstract I now reading this realise) misleading. This is as 'compromise' suggests 'defective' but you showed no evidence of defective function, but rather some fibres have a higher COX activity than other fibres. This is a common observation in muscles from young people also, and we would not call the checkerboard pattern of muscle fibres in the young indicative for fibres with 'compromised' mitochondrial function. It thus also are not 'competent'(line 223) or 'incompetent' fibres, but just fibres with a lower or higher COX activity. However, if you mean to say COX-/SDH^+^ then you are justified, but then you need to clarify that in figure 3f x-axis and identify them as such, and not just COX-.

Thank you for raising this important point. We have now clarified in both the Methods and the Results that the COX intensity status (Pos, Int, Neg) is relative to SDH intensity rather than an estimate of absolute COX intensity. As such, fibers that have low (COXInt) or deficient (COXNeg) intensity relative to SDH have compromised respiratory function since it does not represent low mitochondrial content per se, but rather impaired COX activity relative to SDH. As the reviewer likely knows, this classification comes from the mitochondrial disease literature, where one of our co-authors (Dr. Tanja Taivassalo) is a recognized expert and has specifically advised on this classification strategy.

Page 11 line 233-245: You conclude that the higher mtDNA number in MA than NA is supportive of a lower presence of mtDNA mutations. This is a non-sequitur! You have more mtDNA because you have more mitochondria; this says nothing about mutation load. Please remove this speculation in the opening sentence and as conclusion from this paragraph.

We apologize for our confusing wording. On P. 11, lines 253-255 we now replace the word “protection” with “elevation”, as follows: “parallel elevation of mtDNA copies and mtDNA-encoded proteins”. We hope this clarifies our intended meaning.

Page 12 line 252-266: The altered MICOS may help respiratory capacity in the MA. I accept that. But this is a useful adaptation, and the lower values in NA does not indicate compromised mitochondrial function, but rather that the mt do not need such high cristae density to generate the lower amount of ATP that is needed by the NA than the MA. So, also here, please refrain from using the term 'compromised'.

We agree with your viewpoint here and note that we do not refer to the difference between groups in abundance of these proteins as representing compromise (P. 12, lines 270-273) but rather use the term “compromise” specifically in reference to the impact of experimentally knocking down MICOS components on oxidative phosphorylation function. We have added the word “experimental” prior to “knockdown” to help clarify our meaning. We hope this is satisfactory.

Consider to increase the font in figure 4b; I could only read it when having the figure very close by and straining my eyes.

Done as requested.

Page 14 line 333-341: I don't see the logical link between changes in the spliceosome pathway and mitochondrial function. Please explain why they are, according to you, linked. Circumstantial observations that Mt are up and spliceosome pathways are down in the MA is just that: circumstantial. You can then not say, as that is circular reasoning, that therefore 'a resilience response (that is higher spliceosome) is not required in athletes because of high mt'; that is explaining the observation by the observation.I like the paragraph on page 16, where you disentangle physical activity and ageing.

Several lines of the literature suggest that lower mitochondrial function is associated with upregulation of spliceosome proteins and that the production of alternatively spliced forms of specific proteins counteracts the decline of ATP production that occur with aging in skeletal muscle. Our group recently summarizes this literature in an article published by Nature Aging (Ferrucci, L. et al., The energy–splicing resilience axis hypothesis of aging. Nat Aging (2022) https://doi.org/10.1038/s43587-022-00189-w). In brief, we described that the spliceosome is up-regulated with aging (https://doi.org/10.7554/*eLife*.49874.001) and independent of age is negatively correlated with physical activity even in healthy individuals (https://doi.org/ 10.3389/fphys.2019.00312). Peterson et al., demonstrated that an exercise intervention is followed by a decline in transcripts in skeletal muscle that are related to splicing and RNA processing (https://doi.org/ 10.18632/aging.104096). Based on these findings and a growing literature suggesting that alternative splicing is part of the resilience strategy aimed at improving homeostasis in situations of energetic crisis, we had hypothesized that in this study the spliceosome would have been down-regulated in MA compared to controls. The findings of this study confirm our “a priori” hypothesis. We acknowledge that the connection between mitochondrial function and splicing remain hypothetical and that considerable work need to be done to definitively confirm it. Therefore, we have softened our statement to suggest that our findings “may indicate that such compensatory upregulation of alternative splicing is not required in these individuals because their mitochondrial biology is better protected by other means (e.g., physical activity).”

Page 17: Nice comparison. Do, however, consider to change the term 'affected' by 'higher' or 'lower' where appropriate, as it is that you have measured.

We have replaced “altered” with “under-represented” to make the meaning clear.

Consider the above points also for the discussion. Having said that, the first three paragraphs give a good summary and interpretation of the findings with which I can agree.On page 20 on mitochondria: Please make sure you realise you have not collected evidence of mitochondrial derangement in old age, or prevention thereof in MA, but rather that the abundance of mt proteins was higher. So, in the writing of this part of the discussion do not be tempted to put more into it than you can say based on your data. Note that fewer mitochondria does not mean loss of mitochondrial function; glycolytic fibres in young people -with few mitochondria- function perfectly well!

We appreciate your point here and agree that higher representation of mitochondrial proteins in general would be an expected outcome of higher mitochondrial content. The context in which we are positing that the higher representation of mitochondrial proteins may translate to better mitochondrial function is specifically in the context of proteins that would impact mitochondrial quality control, such as those involved in promoting mitochondrial biogenesis and mitochondrial dynamics, both of which were over-represented in MA vs NA. We summarize our interpretation of this data in lines 570-573, as follows:

“In summary, our data suggest that overrepresentation of mitochondrial quality control proteins and mitochondrial dynamics proteins in octogenarian MA muscle likely translates to better maintenance and remodeling of mitochondrial cristae, with higher energy availability that positively affects cellular adaptation to stress, and better maintenance of muscle metabolism.”

We also point out that our proteomics data and Western blot data found higher representation of oxphos complexes and other cristae proteins relative to outer membrane proteins such as VDAC in MA, which suggests the differences in mitochondria in MA were not just quantitative (more mitochondria) but also qualitative (consistent with a higher cristae density). We hope the reviewer finds this satisfactory.

Page 24 560-563: Probably it is the case that part of the success is related to the genetics of the athletes. However, this may well be a minimal factor, as neither years of training, nor weekly duration of training are related to performance. In addition, there is evidence that people that performance at the age of 80 did not differ significantly between those who had continued competing till at least 85 vs those who had stopped after 80. This observation suggests that there is no selection of the best performers remaining competitive. In addition, again Hepple at al., 1997, but also people as Fiatarone et al., 1990 (training in nonagerians) and Harridge et al., M and N, 1999 show that people even at the age of people in this study can still be trained. I therefore believe that genetics plays only a minor role.The conclusion is justified, though, as I think you will gather by now, I have reservations about this statement in the conclusion that 'that allow sporadic individuals to maintain high levels of physical activity', as the point is that anybody who wants to put in the effort, even in old age, can induce the proteins seen in the MA and as has been shown repeatedly (see refs above) improve their function. So, the main changes observed are a reward for their training, and the therapy suggested is therefore 'train', or at least remain physically active.

Thank you for raising this important point. We wish to underscore that we did in fact identify several proteins (301) that were uniquely over- or under-represented in MA versus NA and which had not been previously identified as exercise-responsive in an analysis by several members of our research group using the same proteomics analysis platform as used in the current investigation. Hence, our data suggest that there are differentially represented proteins in MA that are unlikely to be explained by their exercise habits and speak to the need for considering other (non-exercise) explanations that themselves might confer an advantage to an individual. For example, are there proteomic signatures that have a permissive effect to allow an individual to do more exercise (in other words, high physical function is not exclusively the effect resulting from high physical activity, but rather a high capacity for physical function in some individuals is the cause of their high amounts of physical activity in a permissive sense [they do because they can, not just that they can because they do]). Of course, we also agree completely that exercise remains a highly effective therapy for countering age-related declines in physical function; even if skeletal muscle plasticity typically becomes compromised in advanced age, systemic benefits prevail.